# Identifying effective surveillance measures for swine pathogens using contact networks and mathematical modeling

**Kathleen Moriarty**[1,2]*, **Antoine Champetier**[3], **Francesco Galli**[3], **Salome Dürr**[3], **Nakul Chitnis**[1,2]

**1** University of Basel, Basel, Switzerland, **2** Swiss Tropical and Public Health Institute, Allschwil, Switzerland, **3** Veterinary Public Health Institute, Vetsuisse Faculty, University of Bern, Bern, Switzerland

* kathleen.moriarty@swisstph.ch

**Data availability statement:** Data cannot be shared publicly because of confidentiality

## Abstract

Infectious diseases in livestock have detrimental effects on the health of animals, the livelihood of farmers, and the meat industry. Understanding the specific pathways of disease spread and evaluating the effectiveness of surveillance measures is critical to preventing large outbreaks. Direct livestock transport, transport tours—where a single truck moves livestock between multiple farms in a single journey—and contacts that livestock have with their surrounding environment have been identified as drivers of disease dissemination. The objective of this study was to assess the role of these different pathways in the transmission of several swine pathogens and to evaluate the efficacy of surveillance strategies in identifying outbreaks. To achieve this, we built contact networks for these modes of disease transmission based on empirical data from the Swiss swine production sector. We developed a stochastic, susceptible-infectious-recovered (SIR) type, herd-based model to simulate the spread of multiple pathogens within farms and between farms along the networks. We parameterized the model for Porcine Reproductive and Respiratory Syndrome (PRRS) virus, African Swine Fever (ASF) virus, and *Actinobacillus pleuropneumonia* (APP): three pathogens with distinct clinical patterns, modes of transmission, and contact transmission rates. The model provides insight into the contribution of different contact types to disease dispersion. Our findings highlight that direct truck transport and local spread are the main routes of between-farm transmission. In addition, we analyzed the ability of surveillance measures to detect outbreaks from these distinct pathogens spreading along the contact networks. Farmer-based surveillance programs were the only measures that consistently identified outbreaks of APP and PRRS, and they were able to identify ASF outbreaks almost 8 weeks or more before active slaughterhouse- and network-based surveillance. Our model outcomes give evidence of the prominent transmission pathways and surveillance measures, which could help establish programs to prevent the spread of swine infectious diseases.

concerns due to data belonging to third parties (the pig farmers themselves) and private companies. TVD and AGIS data are available from Identitas AG, Stauffacherstrasse 130A, 3014 Bern, Switzerland, https://www.identitas.ch/, and the Federal Office for Agriculture, 3001 Bern, Switzerland, https://www.blw.admin.ch/de/anwendung-agis, respectively and can be accessed for researchers who meet the criteria for access to confidential data. Requests are treated on an individual basis and signing a data sharing contract will be mandatory.

**Funding:** SD, AC, and FG were supported by the Swiss National Science Foundation (SNSF: https://www.snf.ch/en; project number 182404). The funders had no role in study design, data collection and analysis, decision to publish, or preparation of the manuscript.

**Competing interests:** The authors have declared that no competing interests exist.

## Introduction

Both endemic and epidemic outbreaks of diseases in livestock lead to detrimental loss of animals, economic costs to farmers and a potentially long-term burden on the animal product industry. Three important swine pathogens: African Swine Fever (ASF) virus, Porcine Reproductive and Respiratory Syndrome (PRRS) virus, and *Actinobacillus pleuropneumoniae* (APP), have been shown to significantly affect animal health, farm economics and industry sustainability.

Each pathogen has a distinct clinical impact and mode of transmission. ASF, for example, has caused the death of an estimated 43 million pigs during the first year of the latest outbreak in China [1]. PRRS has been estimated to cost €74,181 per farm per year in Germany [2]. APP also presents a challenge due to growing antimicrobial resistance to veterinary-approved antibiotics, resulting in decreasing treatment efficacy and leading potentially to outbreaks of strains with no form of treatment [3].

Concerns for the loss of animals and the sustainability of the livestock industry have led to extensive research on infectious diseases of swine, providing insight into the modes of disease transmission. In multiple studies, livestock transport has been shown to be an important driver of pathogen dispersion [4–6]. Indirect contacts along transportation routes are established through the sharing of trucks by animals from different farms or truck transport fomites including wheels, undercarriages and drivers. These contacts are also described as sources of pathogen dissemination [7–10]. Additionally, geographic proximity has been shown to facilitate transmission. Spatial modes of dispersal can be explained not only by airborne transmission but also by local veterinary visits, sharing of farming equipment, and wildlife reservoirs and vectors [11–15]. Given the existence of multiple potential transmission pathways, an important task of epidemiological research and modeling is to identify pathways with high transmission risk for each pathogen.

Once key transmission pathways are identified, effective surveillance programs can be proposed and evaluated. For example, surveillance of livestock movements could work well for detecting disease spread through indirect contact, as with foot-and-mouth disease, but could be less effective for pathogens transmitted primarily through direct animal-to-animal contact, such as PRRS. Early detection may prevent large losses of livestock [16]. Identification of effective surveillance strategies is therefore of high relevance. Proposed surveillance programs include the recommended targeted surveillance of large, highly-connected farms [17], the continuous surveillance at slaughterhouses, as currently applied in Switzerland for some pig diseases [18], and the mandatory passive surveillance from farmers and their veterinarians implemented in the EU for several diseases [19]. However, it is unclear which surveillance measures are the most effective for specific pig pathogens with differing transmission dynamics. We hypothesize that pathogens with distinct transmission dynamics result in different performance of surveillance measures.

In order to assess the importance of different contact types and the effectiveness of surveillance, mathematical models have been built to simulate spread of infectious pathogens, in some instances, along contact networks. Many models rely on expert opinion, farmer interviews and averages of the rate and type of contacts calculated from historical data [20–26]. Few models include transport tours [20,27]. The vast majority of models are disease-specific, providing information about the transmission pathways of a single disease such as Foot and Mouth Disease [28], Classical Swine Fever (CSF) [22], or African Swine Fever [23,26,27].

Most models have provided evidence of effective control measures [29,30] and their cost-effectiveness [21]. These models have provided insight into transmission dynamics of livestock pathogens and evidence-based control measures. However, none of the models are implemented with empirical networks of swine transportation, tour, and geographic contacts. In addition, few have evaluated the performance of surveillance measures for multiple pathogens in a comparable fashion.

We developed a data-driven model using empirical data from pig transportation and farm holdings in Switzerland to simulate disease transmission across multiple contact pathways. The swine industry is an important part of the Swiss economy; over 2.8 million piglets are produced each year and an average annual consumption of 21 kg of pork per person [31]. In comparison, the average Swiss consumes 11 kg of beef per person annually [31]. Switzerland records details about all pig holdings, including their location, the type of pig farm, and the average number of pigs. Additionally, Switzerland mandates that all movements of pig herds be recorded. Sterchi et al. [32] have previously described the Swiss pig network of direct transports. These recorded federal data provide a comprehensive database of all pig farms and their herd movements in Switzerland.

The model we developed simulates disease transmission both within farms and between farms via the direct transport, transport tours, and local spread. The within-herd disease dynamics are represented by a stochastic SEIAR (Susceptible, Exposed, Infectious, Asymptomatic, Removed/Recovered) model. For the between-herd disease transmission we incorporated the swine contact network based on the empirical Swiss data from the years 2014 through 2019. We parameterized the model for three important swine pathogens—PRRS virus (PRRSv), ASF virus (ASFv), and APP—which represent distinct transmission characteristics and endemicity levels. Switzerland is classified as free from PRRS, naïve to ASF, and with sporadic outbreaks of APP cases. Fig 1 shows a map of Switzerland, indicating the regions that have had historical cases of APP or PRRS.

In addition, we compared the effect of different contact types on disease transmission. We implemented status-quo and evidence-based surveillance programs. Then we compared the ability of the surveillance programs to identify the pathogen within the Swiss pig population.

The objectives of this study are to assess the role of different contact types in the transmission of multiple swine pathogens and to evaluate the effectiveness of surveillance programs in early outbreak detection. Our model could play an important role in determining the routes of dissemination and the most effective surveillance program for pathogens with distinct transmission routes.

## Materials and methods

### Disease characteristics

PRRS is a viral disease that is characterized by pneumonia, reduced reproduction in sows, and mortality, mainly in piglets. It spreads via contaminated bodily fluids such as urine, serum, saliva, and semen [33,34]. There is strong evidence that the virus can spread within and between swine herds due to direct contact, movement of pigs on contaminated trucks [10], artificial insemination [34], airborne spread [35,36], shared, local suppliers and authorities [13] and wildlife vectors and reservoirs such as infected boar and flies [14,36–38].

For APP, on the other hand, the most common form of transmission is by direct contact of *A. pleuropneumoniae* via nasal secretions. Transmission via fomites is unclear and aerosol is possible, but within short distances [39–41]. There are many strains of APP that circulate in Europe and globally. Strains can be disparate in their clinical presentation and transmission potential. If symptoms develop, they appear within 24 hours after exposure and include

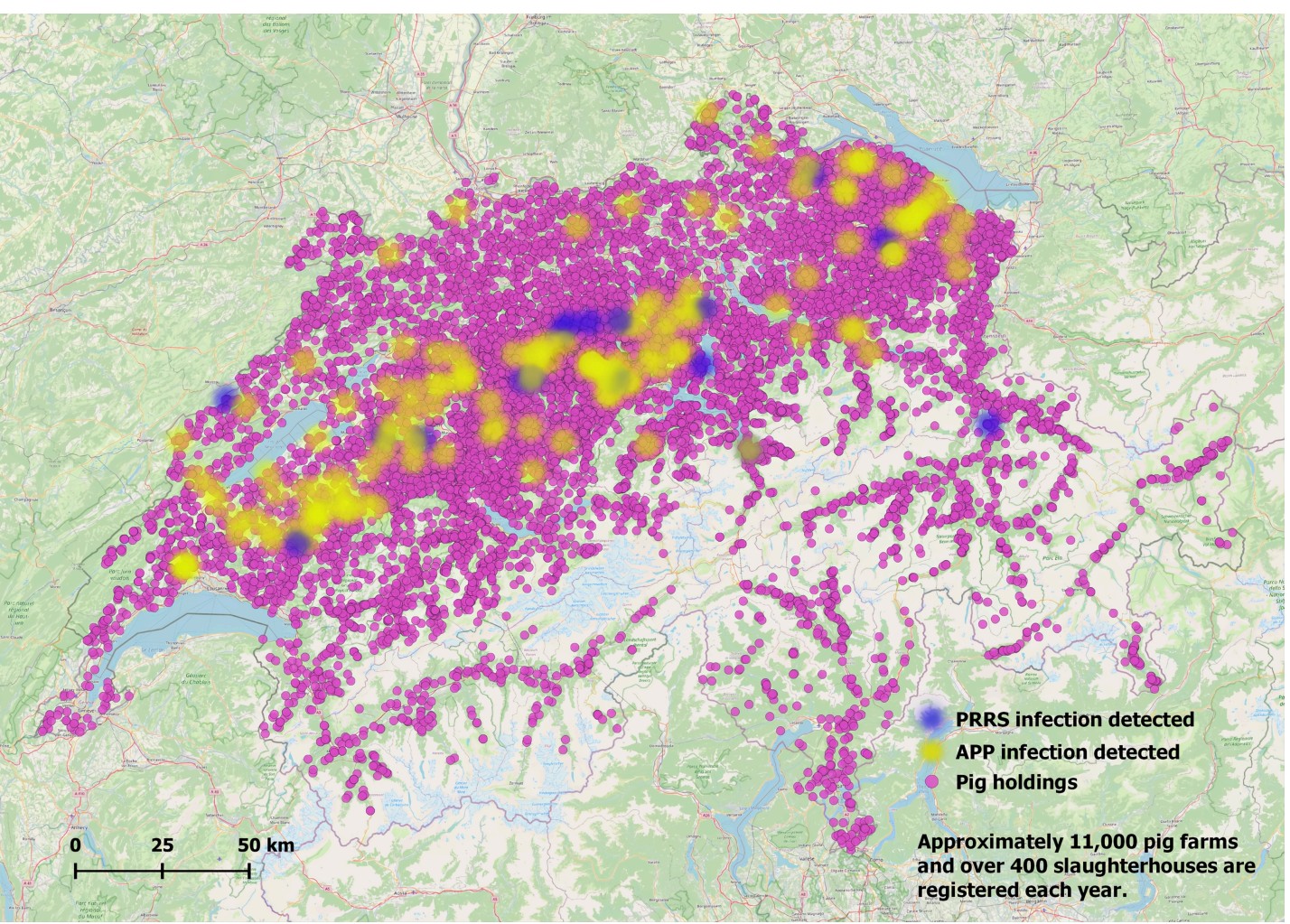

**Fig 1. Map of Switzerland.** Indication of the pig holdings and the regions with APP or PRRS cases between 2014 and 2019.

apathy, anorexia, and vomiting. Pleuropneumonia may develop and lesions on the lungs often form [39].

ASF, similar to APP, has many distinct strains [42]. Most often, it is characterized by moderate to high virulence and high mortality. In symptomatic pigs, fever often develops with apathy. Hemorrhaging and edema are also common. There is evidence that ASFv can be introduced in domestic pig farms via the wild boar [43]. In the outbreaks of the Russian Federation, Oganesyan et al. [44] found that anthropogenic factors were the leading cause of transmission between herds. Besides spread through wildlife, transmission potential exists through contaminated blood, contaminated food and feed, and human contact (e.g., contaminated hands and shoes) [45,46]. Table 1 lists the main features of these diseases.

## Sources of data

We used three different sources of data to build the contact networks: transportation data from the Tierverkehrsdatenbank (TVD) [Animal Tracing Database] provided by Identitas

**Table 1. Disease features.**

| Feature | PRRS | ASF | APP |
|---|---|---|---|
| Clinical Signs | pneumonia, breeding complications in sows, mortality | high fever, high mortality, apathy, hemorrhaging, edema | high fever, apathy, anorexia, vomiting, lesions on the lung |
| Routes of Transmission Between Farms | direct contact, aerosol droplets over long distances, fomites on humans, transport vehicles, farm equipment, artificial insemination[a] | direct contact, wild boar, meat products, blood, fomites on humans, transport vehicles, farm equipment[b] | direct nasal secretion, aerosol droplets over short distances, wild boar[c] |
| Farm Risk Factors of Introduction | geographic proximity to an infected farm, imported semen usage for sows[d] | geographic proximity to an infected farm, located in a municipality with high risk of infected wild boar, feeding of contaminated meat, contaminated fomites[e] | geographic proximity to an infected farm, located in a municipality with high risk of infected wild boar[f] |

[a][33,34], [b][44–46], [c][39–41], [d][14,34–38], [e][43,45,46], [f][39–41].

AG, transportation data from two pig trader companies, and pig farm sizes and characteristics from a pig holding database, Agrarpolitisches Informationssystem (AGIS) [Agricultural Policy Information System], from the Federal Office for Agriculture (FOAG).

**TVD transport data** Daily swine transport data have been collected from the TVD. Switzerland mandates that all movement of pigs must be recorded; therefore each time a pig is transported, it is recorded [47]. We have TVD data from 2014 through 2019, consisting of 1,066,215 direct transports and almost 28,000,000 pig transfers. For pigs, the data are recorded not at the individual animal level, but at the batch level with information of source farm, destination farm, number of pigs and transportation date. A truck often collects and delivers multiple batches, forming a transport tour. The TVD data do not contain information about the sequence of the transports so any intermediary stops in which the pigs have made contacts with other herds are not recorded in the TVD.

We defined the transportation data as a collection of transport records $\mathcal{X}$ where each transport record is a tuple $\mathbf{x} = (s, d, t, n)$, where $s, d \in \mathcal{V}$, $\mathcal{V}$ is the set of all pig holdings, $s$ and $d$ are the source and destination holding, respectively, $t$ is the event date of the transport and $n$ is the number of pigs in the transport.

**Trader data** Two pig trading companies record the truck number, time of collection and time of delivery for each direct transport, which covers 15% of the TVD. With this information, we are able to create the sequence of transports from which we derive the contacts made via the transportation route. Since these two trader companies do not service all farms in Switzerland, we do not have the truck number, collection time, nor delivery time for all records of the TVD data.

We define the trader data as $\mathcal{Y}$ where each transport record is a tuple $\mathbf{y} = (s, d, t, n, k, c, r)$, where $s,d,t,n$ are defined as above, $k$ is the truck number, $c$ is the collection time and $r$ is the delivery time. Since trader transports are a subset of TVD, then $\mathbf{y}$ can be defined as $\mathbf{y} = \mathbf{x} \cup \mathbf{w}$, such that $\mathbf{x} \in \mathcal{X}$ and $\mathbf{w} = (k, c, r)$ for all $\mathbf{y} \in \mathcal{Y}$.

As with the TVD data, we have trader data from 2014 through 2019. With these data, we were able to determine which herds had contacts with other herds via the transport tours as described in Sect Contact Networks.

**AGIS data** Farm characteristics are recorded by the FOAG. The average number of pigs, animal management practices (e.g., access to pasture, indoor stalls only), type of pigs and municipality are stored in the agricultural database. We have AGIS data from 2014 through 2019.

**TVD holding attributes** In addition to the transport data, the TVD also stores holding source and holding destination data. This information includes geographical coordinates and pig holding type, one of 11 categories (e.g., all-year husbandry, breeding, weaning). We connect this information with AGIS data to have a complete dataset of pig holding characteristics. For a limited subset of farms where precise geographic coordinates were not available, we used the centroid of the municipality's polygon as an approximation of the farm location. This allowed us to retain these farms in the analysis while maintaining a reasonable approximation of their location.

We define the pig holding characteristics data as the collection of all pig holdings $\mathcal{Z}$ where each holding record is a tuple $\mathbf{z} = (v, f, a, O, A, b)$, where $v \in \mathcal{V}$, $f$ is the holding type, $a$ is the yearly average number of pigs, $A$ and $O$ are the geographical coordinates of the farm and $b$ is the year.

A description of the data collections and data variables can be found in Table 2.

## Within-farm layer

To simulate the spread of disease within the farm, we built a stochastic susceptible-exposed-infectious-asymptomatic-recovered (SEIAR) model (Fig 2, Tables 3 and 4). We used the $\tau$-leap method with Poisson and binomial sampling to model stochasticity [48]. $\tau$, in this case, is set to 1 day because the transportation of pigs is recorded on a daily basis. Since we are interested in short-term dynamics, such as date of first detection, natural birth and death can be ignored. In addition, we assumed homogeneous mixing of pigs at the farm with frequency-dependent transmission. Although density-dependent transmission has often been used for animal diseases, this is more appropriate for animals kept in open spaces, such as on pasture, where densities can change. Pig farms in Switzerland are strictly regulated with a clearly defined maximum number of animals per space by legislation. This leads to a maximum number of pigs per barn, limiting the number of contacts for each animal, even if the number of pigs per farm

**Table 2. Data set and variable definitions.**

| Data Sets | |
|---|---|
| **Variable** | **Description** |
| $\mathcal{X}$ | Collection of transport records |
| $\mathcal{V}$ | Set of pig holdings |
| $\mathcal{Z}$ | Collection of pig holdings with characteristics |
| $\mathcal{Y}$ | Collection of transport records from the trader data |
| **Data Variables** | |
| **Variable** | **Description** |
| $s$ | Source holding from direct transport data |
| $d$ | Destination holding from direct transport data |
| $t$ | Event date from direct transport data |
| $n$ | Number of pigs from $s$ to $d$ from direct transport data |
| $k$ | Truck number from trader data |
| $c$ | Loading time from trader data |
| $r$ | Unloading time from trader data |
| $v$ | Pig holding id |
| $f$ | Pig holding type |
| $a$ | Pig holding average number of pigs for the year |
| $A$ | Pig holding Easting |
| $O$ | Pig holding Northing |
| $b$ | Pig holding record year |

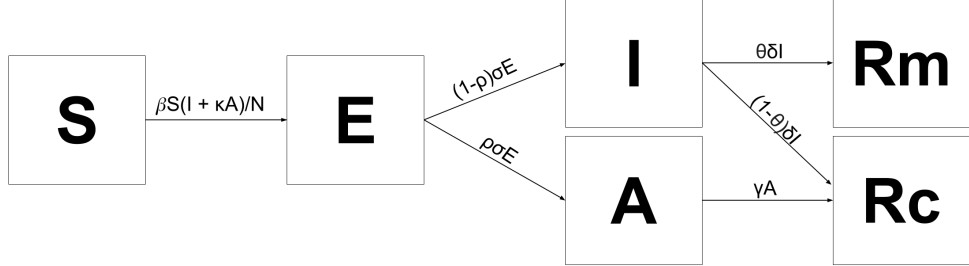

**Fig 2. Schematic of the SEIAR model showing movement rates between compartments.** Model state variables are shown in Table 3 and parameters in Table 4.

**Table 3. State variables.**

| Variable | Description |
|---|---|
| S | Number of Susceptible Pigs on the Farm |
| E | Number of Exposed Pigs on the Farm |
| I | Number of Infectious Pigs with Symptoms on the Farm |
| A | Number of Infectious Pigs without Symptoms on the Farm |
| Rm | Number of Removed Pigs on the Farm |
| Rc | Number of Recovered Pigs on the Farm |

increases. Hence a frequency-dependent transmission assumption is more appropriate across farm systems in Switzerland.

There are six possible events at each time step. A susceptible pig is exposed, entering the latency phase, ($S \rightarrow E$), a latent pig moves to either the infectious symptomatic ($E \rightarrow I$) or infectious asymptomatic ($E \rightarrow A$) state, an infectious symptomatic pig is removed ($I \rightarrow Rm$) or recovered ($I \rightarrow Rc$) or an asymptomatic pig recovers ($A \rightarrow Rc$). The updated compartments at time $t$ are represented as:

$$S_t = S_{t-1} - P_{SE}, \tag{1a}$$

$$E_t = E_{t-1} + P_{SE} - P_{EI} - P_{EA}, \tag{1b}$$

$$I_t = I_{t-1} + P_{EI} - P_{IRm} - P_{IRc}, \tag{1c}$$

$$A_t = A_{t-1} + P_{EA} - P_{ARc}, \tag{1d}$$

$$Rm_t = Rm_{t-1} + P_{IRm}, \tag{1e}$$

$$Rc_t = Rc_{t-1} + P_{ARc} + P_{IRc}, \tag{1f}$$

where the probability events at time $t$ are described as follows.

$$\tilde{P}_{SE} \sim \text{Poisson}\left(\tau \times \beta \times S_{t-1} \times \frac{I_{t-1} + \kappa \times A_{t-1}}{N}\right), \tag{2a}$$

$$\text{where } P_{SE} = \min(\tilde{P}_{SE}, S_{t-1}) \tag{2b}$$

$$P_{EIA} \sim \text{Bi}\left(E_{t-1}, 1 - e^{-\sigma\tau}\right) \tag{2c}$$

$$P_{EA} \sim \text{Bi}\left(P_{EIA}, \rho\right) \tag{2d}$$

$$P_{EI} = P_{EIA} - P_{EA} \tag{2e}$$

**Table 4. Within-farm model parameters.**

| Description | Variable | PRRS | ASF | APP | Units | Source |
|---|---|---|---|---|---|---|
| Contact transmission rate | $\beta$ | 0.054 | 1.75 | 0.27 | days$^{-1}$ | [PRRS] [49] [ASF]$^a$ [APP] [50] |
| Factor of infectivity rate for asymptomatic pigs | $\kappa$ | 0.25$^b$ | 0.15 | 0.37$^c$ | - | [PRRS] assumed [ASF] [51,52] [APP] [50] |
| Average duration in exposed state | $\frac{1}{\sigma}$ | 1 | 6.93 | 1 | days | [PRRS] [53] [ASF]$^d$ [APP] [54] |
| Proportion of pigs infected that are asymptomatic | $\rho$ | 0 or 1$^e$ | 0.10 | 0.50 | - | [PRRS] [49] [ASF]$^f$ [APP] [55] |
| Average duration in infectious state | $\frac{1}{\delta}$ | 56 | 6.97 | 52 | days | [PRRS] [49] [ASF]$^g$ [APP] [56] |
| Prop of pigs in infectious state that die due to disease | $\theta$ | 0.20 | 0.95 | 0.05 | - | [PRRS] [57] [ASF] [51,52] [APP] [55] |
| Average duration in infectious state of asymptomatic pigs | $\frac{1}{\gamma}$ | 56 | 40 | 52 | days | [PRRS] [49] [ASF] [51,52] [APP] [56] |
| Number of pigs on the farm | $N$ | - | - | - | - | Initialized as $a$ (defined in Table 2) |
| Reproductive number | $R_0$ | 3 or 0.76$^e$ | 12 | 9.6 | - | Calculated from above parameters and explained in Sect Parameters |

$^a$[58,59], $^b$ Knowing that asymptomatic pigs are less infectious than symptomatic pigs for PRRS, we assumed the factor to be 0.25. $^c\left(\frac{0.1}{\beta_{APP}}\right)$, $^d$[59,60], $^e$Since the clinical presentation of PRRS is different among sows and piglets than for fattening pigs, we adapt our model to reflect this. Farms with sows have no asymptomatic cases while farms without are all asymptomatic, $^f$[51,52,61], $^g$[45,58–60]

$$P_{IR} \sim \text{Bi}\left(I_{t-1}, 1 - e^{-\delta\tau}\right) \tag{2f}$$

$$P_{IRm} \sim \text{Bi}\left(P_{IR}, \theta\right) \tag{2g}$$

$$P_{IRc} = P_{IR} - P_{IRm} \tag{2h}$$

$$P_{ARc} \sim \text{Bi}\left(A_{t-1}, 1 - e^{-\gamma\tau}\right), \tag{2i}$$

where $P_{xy}$ are the number of pigs moving from compartment $x$ to compartment $y$, $\tau$ is the time step taken, in this case 1 day, $\beta$ is the contact transmission rate, $\kappa$ is the factor of infectivity rate for the asymptomatic, $\sigma$ is the rate at which an exposed pig moves to infectious state, $\rho$ is the proportion of the infectious that are asymptomatic, $\theta$ is the proportion of pigs that do not survive the infectious state, $\delta$ is the rate at which infectious symptomatic are either removed or recovered and $\gamma$ is the recovery rate for infectious asymptomatic. For $P_{SE}$, if the draw from the Poisson distribution was greater than $S_{t-1}$, then $S_{t-1}$ was used.

## Contact networks

To spread the disease between herds, we need to know which contacts one infected herd has had with other herds. Using the transport data, the tour data, and the geographic coordinates of the farms, we built a dynamic, directed network in which each farm **v** is a node and the edges $\mathcal{E}$ connecting the nodes are drawn when a contact exists between the farms. The edge

exists only for one day because the transport and tour data are temporal; each day a new network is created. The network can be described as a graph, $G = (\mathcal{V}, \mathcal{E})$, where $\mathcal{V}$ is the set of all nodes (pig holdings) and $\mathcal{E}$ is the set of all edges. Since we have different types of contacts with varying probability of pathogen transmission, each edge is labeled based on the contact type. Edges are labeled by the function $\ell$ where $\ell(.)$ assigns a label to the edge and is one of $D,P,I,T,G$ (Table 5). Two nodes can have multiple edges connecting them for a given day.

**Direct Transport Edges** Following our notation from the direct transport data of the TVD, we create an edge for all $\mathbf{x} \in \mathcal{X}$ such that $\ell(e_{x_s x_d}) = $ 'D', for direct transport.

**Tour Edges** With the trader data, we distinguished three levels of contact type: direct pig contact via truck sharing, indirect pig contact via truck sharing (when herds share the same truck but are not on the truck at the same time), and indirect contact from fomites on the exterior of the truck and truck driver.

To illustrate a direct pig contact via truck sharing, suppose a truck first picks up a herd from farm $s_i$ then picks up another herd from $s_j$. The truck proceeds to drop off the herd from $s_j$ at $d_j$ then drops off the herd from $s_i$ at $d_i$. A diagram of this example is shown in Fig 3a. From our trader data, we have $y_i, y_j \in \mathcal{Y}$, and

$$\mathbf{y_i} = (s_i, d_i, t_i, n_i, k_i, c_i, r_i) \text{ and}$$
$$\mathbf{y_j} = (s_j, d_j, t_j, n_j, k_j, c_j, r_j),$$

where $t_i = t_j$, $k_i = k_j$ and with the collection and delivery times as $c_i < c_j < r_j < r_i$.

Since the pigs from $s_i$ share the tour truck with the pigs that end at $d_j$, there is direct pig contact via truck sharing. Hence, we create an edge labeled $\ell(e_{s_i d_j}) = $ 'P'. The same is true for $s_j$ and $d_i$ with edge $\ell(e_{s_j d_i}) = $ 'P'.

Additionally, a contact between $s_i$ and $s_j$ via exterior truck fomites and truck driver exists. This leads to an edge that is labeled $\ell(e_{s_i s_j}) = $ 'T'.

A list of all edges created from this transport are listed in Fig 3a.

An example of an indirect pig contact via truck sharing (when batches share the same truck but are not on the truck at the same time) can be illustrated with another common scenario (Fig 3b). Suppose pigs from farm $s_k$ are taken to farm $d_k$, then without cleaning the truck, pigs from farm $s_m$ are taken to farm $d_m$. Our records would show

$$\mathbf{y_k} = (s_k, d_k, t_k, n_k, k_k, c_k, r_k) \text{ and}$$
$$\mathbf{y_m} = (s_m, d_m, t_m, n_m, k_m, c_m, r_m),$$

where $t_k = t_m$, $k_k = k_m$ and with $c_k < r_k < c_m < r_m$. Since the herd from $s_m$ spends time on the truck after the herd from $s_k$, there is indirect truck sharing contact leading to the edge labeled $\ell(e_{s_k d_m}) = $ 'I'. The other edges formed from this transport are listed in the caption of the figure.

**Table 5. Edge label descriptions.**

| Edge Label | Edge Type | Description |
|---|---|---|
| D | Direct Transport | Direct pig transport |
| P | Tour | Direct pig contact via truck sharing |
| I | Tour | Indirect pig contact via truck sharing |
| T | Tour | Fomites via truck driver and exterior truck equipment |
| G | Geographic | Local pathogen dispersion |

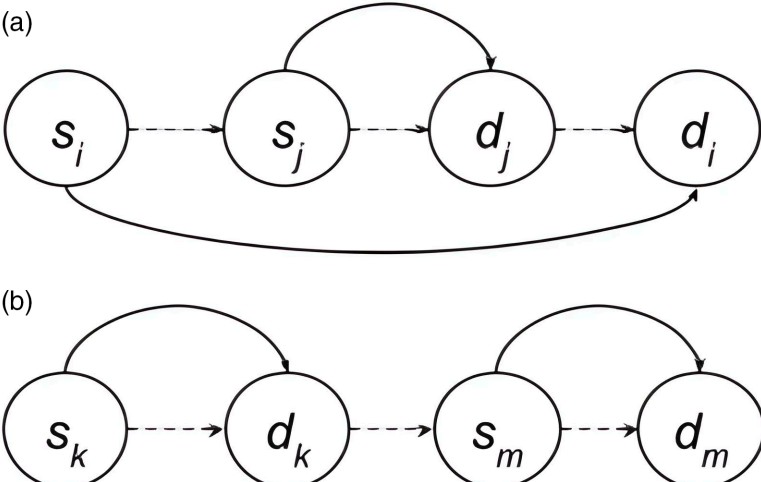

**Fig 3. Diagrams for two scenarios of truck tours. (a)** Solid line indicates direct pig movement. **(b)** dashed line indicates truck temporal movement.

For transport edges, if farms had multiple contact types on the same day, we chose the type that had the highest probability for disease transmission. The order of highest to lowest probability was D, P, I, T. Although not shown here, the same holding can be both a source and a destination for different transports.

**Geographic Edges** For the geographic network, the geographic coordinates, $A$ and $O$ from the TVD data, were used to create edges between farms that were in close proximity. We created edges between farms based on Euclidean distance. The edges are defined as $\ell(e_{sd})$ = 'G' where $s, d \in \mathcal{V}$ and $\text{dist}(s, d) < k$. The function $\text{dist}()$ is the Euclidean distance in kilometers between $s$ and $d$. We calculated this distance using the open source software QGIS version 3.20.

Unlike the transport edges, the geographic-based edges are bidirectional, whenever there is an edge $\ell(e_{sd})$ = 'G', then there exists an edge $\ell(e_{ds})$ = 'G'.

**Example** To clarify these different contact types, we illustrate the network described in the Tour Edges paragraph of Sect Contact Networks with the additional assumption that $\text{dist}(s_i, d_i) < k$, $\text{dist}(s_j, s_k) < k$ and $\text{dist}(s_k, s_n) < k$.

The network for this farm contact scenario is drawn in Fig 4.

## Between farm layer

**Spread via Direct Transport** At time $t$, if a farm has infected pigs and if there is a transport of pigs from the farm, then there is probability for infected pigs to be transported to the destination farm. This probability is based on the number of pigs transported ($P_N$) and the proportion of infected pigs on the farm. The number of exposed ($P_{hE}$), infectious symptomatic ($P_{hI}$), and infectious asymptomatic ($P_{hA}$) pigs in the transport is represented by

$$\tilde{P}_{hE} \sim \text{Poisson}\left(P_N \times \frac{E_{st}}{N_s}\right), \text{ where } P_{hE} = \min(\tilde{P}_{hE}, P_N, E_{st}), \tag{3a}$$

$$\tilde{P}_{hI} \sim \text{Poisson}\left(P_N \times \frac{I_{st}}{N_s}\right), \text{ where } P_{hI} = \min(\tilde{P}_{hI}, P_N - P_{hE}, I_{st}), \tag{3b}$$

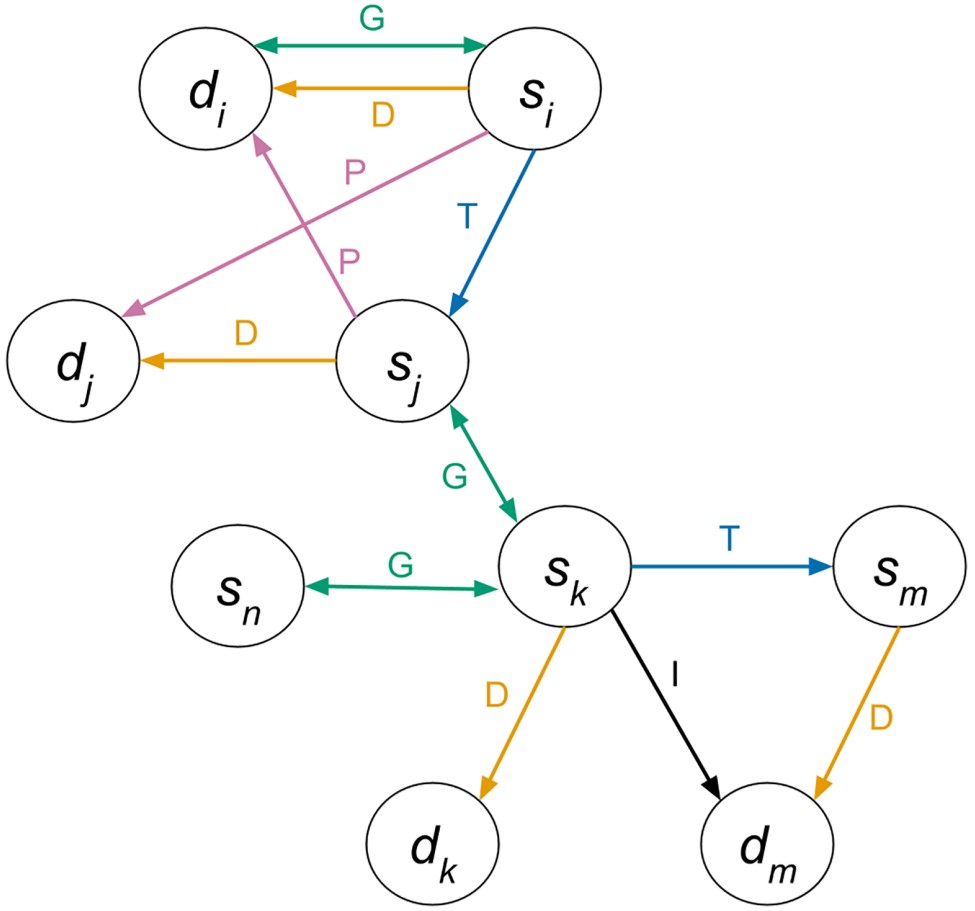

**Fig 4. Example contact network for one day.**

$$\tilde{P}_{hA} \sim \text{Poisson}\left(P_N \times \frac{A_{st}}{N_s}\right), \text{ where } P_{hA} = \min(\tilde{P}_{hA}, P_N - P_{hE} - P_{hI}, A_{st}), \tag{3c}$$

where $P_N$ is the number of pigs transported from the farm, $E_{st}$ is the number of exposed pigs on source farm $s$ at time $t$, $I_{st}$ is the number infectious symptomatic pigs on source farm $s$ at time $t$, $A_{st}$ is the number infectious asymptomatic pigs on source farm $s$ at time $t$, $N_s$ is the total number of pigs on farm $s$ and $\tau$ is the time step as defined above. If $\tilde{P}_{hE}$, $\tilde{P}_{hI}$ or $\tilde{P}_{hA}$ is greater than $P_N$, then $P_N$ is used.

If $P_{hE}$, $P_{hI}$ or $P_{hA}$ is greater than 0, then the following compartments are updated for the source and destination farms.

$$E_{st} = E_{s(t-1)} - P_{hE}, \tag{4a}$$

$$I_{st} = I_{s(t-1)} - P_{hI}, \tag{4b}$$

$$A_{st} = A_{s(t-1)} - P_{hA}, \tag{4c}$$

$$E_{dt} = E_{d(t-1)} + P_{hE}, \tag{4d}$$

$$I_{dt} = I_{d(t-1)} + P_{hI}, \tag{4e}$$

$$A_{dt} = A_{d(t-1)} + P_{hA}, \tag{4f}$$

where $E_{dt}$, $I_{dt}$, $A_{dt}$ are the numbers of exposed, infectious symptomatic, infectious asymptomatic, respectively, pigs on the destination farm $d$ at time $t$, and all other values are as explained earlier.

**Spread via Tour** If at least one infected pig is chosen for the transport, then the tour network is scanned for additional herd contacts. The numbers of pigs infected from the tours are represented as:

$$P_P \sim \text{Poisson}\left(\phi \times S_p \times (P_{hI} + \kappa \times P_{hA})\right) \tag{5a}$$

$$P_I \sim \text{Poisson}\left(\psi \times S_p \times (P_{hI} + \kappa \times P_{hA})\right) \tag{5b}$$

$$\tilde{P}_T \sim \text{Poisson}\left(\eta \times S_{dt} \times (P_{hI} + \kappa \times P_{hA})\right), \text{ where } P_T = \min(\tilde{P}_T, S_{dt}), \tag{5c}$$

where $P_P$ is the number of pigs infected via direct truck sharing, $P_I$ via indirect truck sharing, and $P_T$ via exterior truck fomites. $\phi$, $\psi$, and $\eta$ are transmission rates for each contact category. $S_p$ is the number of susceptible pigs transported to the contact destination farm. $P_{hI}$ is the number of infected symptomatic pigs on the truck from the source farm, $P_{hA}$ is the number of infected asymptomatic pigs on the truck from the source farm, and $S_{dt}$ is the number of susceptible pigs on the contact destination farm at time t. For $P_T$, if the draw is greater than $S_{dt}$, then $S_{dt}$ is used.

If $P_P$, $P_I$ or $P_T$ is greater than 0, then the following compartments are updated for the source and destination farms, respectively.

$$E_{dt} = E_{d(t-1)} + P_P, \tag{6a}$$

$$E_{dt} = E_{d(t-1)} + P_I, \tag{6b}$$

$$S_{dt} = S_{d(t-1)} - P_T, \tag{6c}$$

$$E_{dt} = E_{d(t-1)} + P_T, \tag{6d}$$

The parameters chosen for the model simulation are explained in Sect Parameters.

**Spread via Geographic Proximity** If an infected farm is within $k$ kilometers of other pig farms, then the daily number of susceptible pigs infected in those farms is modeled as

$$\tilde{P}_g \sim \text{Poisson}\left(\tau \times \omega \times S_{dt} \times (I_{st} + \kappa \times A_{st})\right), \text{ where } P_g = \min(\tilde{P}_g, S_{dt}), \tag{7a}$$

where all the variables are as mentioned above, $P_g$ is the number of pigs infected via geographic proximity and $\omega$ is the geographic transmission rate with values explained in Sect Parameters. If the draw from the Poisson distribution is greater than $S_{dt}$, then $S_{dt}$ is used.

If $P_g$ is greater than 0, then the following compartments are updated for the destination farm,

$$S_{dt} = S_{d(t-1)} - P_g \tag{8a}$$

$$E_{dt} = E_{d(t-1)} + P_g. \tag{8b}$$

## Index case

The index case is randomly chosen from all farms, with farms weighted according to their disease-specific introduction potential, based on the probabilities and risk factors shown in Table 1. Farms with the lowest likelihood of being the index case are assigned a weight of 1,

and the weight doubles for each increase in risk. The weights are shown in Table B and Table C in S1 Tables.

**PRRS** For PRRS, a likely introduction of PRRS virus (PRRSv) into the Swiss system is through boar semen as was the case in the latest outbreak in 2013 [34,62]. Therefore, more weight is given to farms with sows and even more weight given to farms with sows that do not have boars, increasing the likelihood of importing semen from infected boars. We used the AGIS data with information of farm pig type to identify the higher-risk farms, according to this rational. Additionally, there is an increased risk of introduction across borders with aerial spread and local interaction at the Swiss borders where PRRS is still persistent. For that reason, farms at the borders of Switzerland are given more weight. Using QGIS version 3.20 and municipality maps from the Federal Office of Topography, we were able to create a list of Swiss municipalities that shared a border with a neighboring country. Finally, access to open pastures increases the risk of PRRSv introduction due to the aerial transmission and wildlife reservoirs. The AGIS data contain outdoor access level labeled from 1 to 4, where 4 indicates access to open pastures and 1 indicates only indoors. Table B in S1 Tables shows the weighting scheme.

**ASF** Since it is assumed that ASF introduction will be due to wild boar [63], a higher probability was given to farms located in cantons in which disease introduction of wild boar was most likely. Amado et al. [64] calculated this canton risk by first determining the relative abundance of the wild boar in Switzerland and then determining the points of human and wild boar contact via motorways and rest stops. Higher probability was also given to farms in which the pigs have access to open pastures compared to those that remain indoors. Table C in S1 Tables provides the details.

**APP** Reiner et al. [65] found 35.8% of wild boar in Germany to be infected with APP. Because of this risk of transmission from wild boar, APP index case weighting follows the same mechanism as ASF.

## Parameters

**Within-herd transmission** The model parameters for the within-herd layer of PRRS and ASF were chosen based on published, peer-reviewed literature of experimental field studies. These parameters are listed in Table 6. A common, summarizing parameter for transmissible pathogens is the reproductive number, $R_0$. It represents the number of pigs to which a single infected pig will transmit the pathogen. To calculate $R_0$, we multiply the daily contact transmission rate by the number of days that the pig will be infectious. This is different for asymptomatic and symptomatic pigs. Using the parameter variables described in Table 6, the expressions for $R_0$ are

$$R_0^{(\text{symptomatic})} = \beta \times \frac{1}{\delta}, \tag{9a}$$

$$R_0^{(\text{asymptomatic})} = \beta \times \kappa \times \frac{1}{\gamma}, \tag{9b}$$

$$R_0^{(\text{overall})} = (1 - \rho) \times \beta \times \frac{1}{\delta} + \rho \times \beta \times \kappa \times \frac{1}{\gamma}. \tag{9c}$$

Based on the within-herd parameters from the literature, we can calculate the simulated within-herd $R_0$. For PRRS, in many studies, evidence suggests that sows have a high chance of passing the virus onto their offspring; there is a high contagiousness of piglets, and older pigs

**Table 6. Between farm parameters.**

| Description | Variable | CSF[a] | PRRS[b] | ASF[c] | APP | Units |
|---|---|---|---|---|---|---|
| Pig direct truck sharing transmission rate | $\phi$ | $2.6 \times 10^{-3}$ | $3.7 \times 10^{-5}$ | $1.2 \times 10^{-3}$ | 0.1 | transport$^{-1}$ |
| Pig indirect truck sharing transmission rate | $\psi$ | $1.3 \times 10^{-3}$ | $2.8 \times 10^{-5}$ | $6 \times 10^{-4}$ | 0.01 | transport$^{-1}$ |
| Exterior truck fomite transmission rate | $\eta$ | $4.2 \times 10^{-4}$ | $9 \times 10^{-6}$ | $1.9 \times 10^{-4}$ | 0 | transport$^{-1}$ |
| Geographic distance of within 2 km transmission rate | $\omega$ | $5.8 \times 10^{-5}$ | $8.3 \times 10^{-7}$ | $2.7 \times 10^{-5}$ | 0 | day$^{-1}$ |

[a]The equation to evaluate the $\psi$ parameter for CSF is $\psi_{CSF} = \frac{\phi_{CSF}}{2}$

[b]The equations to evaluate the parameters for PRRS are

$\phi_{PRRS} = \frac{\beta_{PRRS}}{\beta_{CSF}} \times \phi_{CSF}$

$\psi_{PRRS} = \left(\frac{\psi_{CSF}}{\phi_{CSF}} \times 1.5\right) \times \phi_{PRRS}$

$\eta_{PRRS} = \left(\frac{\eta_{CSF}}{\phi_{CSF}} \times 1.5\right) \times \phi_{PRRS}$

$\omega_{PRRS} = \left(\frac{\omega_{CSF}}{\phi_{CSF}}\right) \times \phi_{PRRS}$

[c]The equations to evaluate the parameters for ASF are

$\phi_{ASF} = \frac{\beta_{ASF}}{\beta_{CSF}} \times \phi_{CSF}$

$\psi_{ASF} = \left(\frac{\psi_{CSF}}{\phi_{CSF}}\right) \times \phi_{ASF}$

$\eta_{ASF} = \left(\frac{\eta_{CSF}}{\phi_{CSF}}\right) \times \phi_{ASF}$

$\omega_{ASF} = \left(\frac{\omega_{CSF}}{\phi_{CSF}}\right) \times \phi_{ASF}$

often develop an immune response [66]. For these reasons, we have modeled distinct transmission patterns for farms with sows and farms without sows. $R_0$ for farms with sows is simulated at 3 and is supported by the findings of [49]. Farms without sows have an $R_0$ of 0.76; this is supported by multiple studies that have found $R_0$ less than 1 for PRRS when the pigs have increased immunity [67–70].

The simulated $R_0$ value for within-herd spread of ASF is 12. This is consistent with what other studies have found [58,59].

To determine the within-herd transmission parameters for APP, we had to consider the varying infectivity levels that have been characteristic of APP infection. On days of high colony count, it has been found that the contact transmission rate, $\beta$, increases by tenfold [71]. Velthuis et al. [50] found $\beta$ to be 0.1 for the period of low infectivity. So for the high infectivity phase, $\beta$ would be 1. Sassu et al. [54] found the acute phase to be 6-10 days; we chose 10 days and assumed that this was the period of high infectivity. The duration of infection was approximated to be 52 days [56]. To determine the average $\beta$ for symptomatic pigs, we weighted the two $\beta$ values (1, 0.1) across the expected duration (10 days, and 42 days) which resulted in 0.27. We used this for the APP contact transmission rate.

**Between-herd transmission** To our knowledge, there is no published study that has determined the transport and local transmission parameters of PRRS or ASF. In order to estimate these parameters, we derived them from Classical Swine Fever virus (CSFv), a similar infectious viral pathogen of swine. Stegeman et al. [72] quantified the direct and indirect transmission rates for CSF from the 1998 outbreak in The Netherlands. The differences between CSF, ASF, and PRRS were based on the within-herd transmission rates, evidence from pathway importance in field studies, and expert opinion.

Stegeman et al. [72] estimated the transmission rate of one infected herd to a susceptible herd via contact in a transport truck to be 0.067. We used this rate and divided it by the average number of pigs in the susceptible herd (26) to determine the transmission rate of 1 infected herd to 1 susceptible pig. Unfortunately, they did not estimate the rate of transmission via indirect truck-sharing, but because of the lower likelihood of passing the virus via

indirect contact, we make the simplifying assumption that the indirect truck-sharing rate is half the direct truck-sharing rate. For the exterior truck fomite transmission rate, they estimated 0.011 per contact. We followed the same logic as with the direct truck-sharing by dividing by the average number of susceptible pigs in the herd (26). As for the local, geographic transmission, Stegeman et al. [72] considered both aerial spread and local contacts. For the geographical distance, we averaged their estimated 0–500 meters (0.027), 500m–1km (0.0078), and 1km–2km (0.00006) weekly rates and converted them into a single daily rate. We used their local person and equipment sharing contact transmission rate (0.0068) and assumed that there was one local contact per day [73]. We summed the geographic spread and local contact rates to form the aggregate parameter of local spread of CSF. Then we divided by the average number of susceptible pigs on the farm (145) to obtain the 1 infected herd to 1 infected pig rate. To be consistent with the Stegeman et al. [72] study, we chose a geographic spread threshold ($k$) as described in Sect Between Farm Layer to be 2 kilometers.

For the PRRS and ASF parameters, we first derived the direct truck-sharing parameter ($\phi$). To do this, we calculated the ratio of the contact transmission rate ($\beta$) to the direct truck-sharing rate ($\phi$) of CSF. From previous studies of CSF, it has been shown that the $\beta$ within a farm is on average 3.76 [74]. We then assume that this ratio (of $\beta$ for CSF to $\phi$ for CSF) is equal to the ratio of $\beta$ to $\phi$ for PRRS and ASF.

For the indirect truck-sharing parameters ($\psi$), the pathway of transmission is predominantly through urine and feces. Several studies have concluded that nasal and rectal secretions are not significant transmission drivers for either CSFv or ASFv [45,75]. For ASF, we use the same ratio of $\psi$ to $\phi$ as for CSF with evidence of ASF transmission through indirect contact in recent outbreaks in Estonia [76]. PRRSv has been shown to spread easily through urine, nasal, and feces secretion and through truck fomites [10,77]. We therefore chose to increase the ratio of indirect truck-sharing rate, $\psi$, to $\phi$ for PRSS as compared to the ratio of $\psi$ to $\phi$ for CSF to account for this increased rate of transmission. In the absence of concrete data on the relative ratio of $\psi$ to $\phi$ for PRRS, we make the arbitrary choice of using 1.5 as the inflation factor. In S1 Appendix, Section A and Section B, we share additional data on sensitivity analyses to consider the impact of this choice.

We similarly assume that the ratio of transmission by fomites ($\eta$) to transmission during direct transfer ($\phi$) is the same for ASF and CSF; and that the ratio of $\eta$ to $\phi$ for PRRS is 1.5 times higher than the ratio of $\eta$ to $\phi$ for CSF [78]. Since the base transmission rate, $\beta$, is substantially higher for ASF than PRRS, the transmission rate through fomites is higher for ASF than PRRS, despite the increased inflation for PRRS. This is corroborated by findings from ASF outbreaks in which between-farm spread was due to human shoes and clothing [44,79].

From previous studies, local spread ($\omega$) has been found to be an important factor for CSFv, [80], PRRSv [14] and ASFv, mostly through the persistence in wild boar [43]. We therefore used the same ratio of $\omega$ to $\phi$ for ASF and PRRS as for CSF.

The base rates of CSF and the calculated rates for PRRS and ASF, as well as all equations relating them, are displayed in Table 6.

To estimate the between-herd parameters for APP, we gathered information from published field studies. Tobias et al. [81] found that the transmission rate by direct pig contact was 10 times higher than indirect contact. The direct contact transmission number was estimated to be 0.1 per day for direct truck sharing [50]. We assumed the indirect truck sharing rate was thus 0.01. APP has been shown to not survive long in the environment and therefore not a concern for indirect pig transmission via exterior truck contamination [40]. Kristensen et al. [82] found aerial spread rare under field experiments of 1 meter. For these two reasons, we set APP exterior truck fomite transmission rate and geographic distance transmission rate to 0.

## Surveillance programs and control measures

One of the main goals of this project was to compare surveillance programs among diverse pathogens. We chose three types of surveillance programs to implement: farmer-based surveillance, slaughterhouse surveillance, and network metric-based surveillance.

**Farmer-based surveillance** To support farmers when unusual health problems arise within their herd, the Swiss Federal Office for Food Safety and Veterinary Services (FSVO) subsidizes diagnostic autopsy of up to three pigs for clarification [83]. The farmers, together with their veterinarians, can utilize the program when there is an increase in morbidity, attrition, unusual symptoms, or recurring problems of unknown cause. We incorporated two scenarios for which a farmer would initiate the surveillance: a morbidity scenario where a certain percent of the herd shows clinical signs and a mortality scenario where a certain percent increase in mortality is reached. For each scenario, we chose three different thresholds: for the percent morbidity, surveillance was initiated after 20%, 30% and 40% of the herd showed clinical signs (proportion of pigs within a farm in the infectious symptomatic ($I$) class) and for the mortality scenario, we used an increase of 5, 10, and 15% in baseline mortality. Grosse et al. [84] found that the average mortality on German fattening farms was 2.56% while Ruckli et al. [85] found that across 63 different European farms, the average mortality of weaners and finishers was 2.43%. We consider 2.5% as the baseline mortality. The surveillance program is initiated once the disease-induced mortality reaches 7.56, 12.56 and 17.56% of the herd. Once the farm met these thresholds, a delay of 10 days was included before testing to account for the time required for veterinarians to visit and for the carcasses or samples to be analyzed. Because there is a risk that farmers will not initiate surveillance when there is an increase of morbidity and mortality [86], we chose to compare farmer participation levels. For both scenarios and ranges in threshold values, we compared the effectiveness of the program based on 100%, 90%, and 60% participation. The 10% and 40% of farmers who would not initiate surveillance were randomly chosen for each iteration.

**Slaughter-based surveillance** We implemented this scenario based on the slaughterhouse surveillance program currently implemented in Switzerland to detect outbreaks of PRRS and Aujeszky's disease. The program begins by first enlisting the nine largest slaughterhouses. Then, pigs are tested upon arrival, selecting at least 6 pigs from each source farm. The program begins on the first day of the year and continues until a set number of herds is reached. The total number of farms tested at each slaughterhouse is dependent on the number of infected herds from the previous year. In 2020, the program tested 7,200 pigs from 9 slaughterhouses [18]; this is an average of 134 herds tested at each of the 9 slaughterhouses.

We implemented this slaughterhouse surveillance program in the model by selecting the 9 slaughterhouses with the largest number of incoming transports (to approximate the 9 largest slaughterhouses). Then we tested 6 pigs from 134 herds when they arrived at the slaughterhouse, with surveillance beginning at the start of the simulation when the index case was introduced into the Swiss swine network. If the pigs were infectious symptomatic ($I$) or infectious asymptomatic ($A$), they were identified as a positive case.

In addition to the surveillance program with the 9 selected slaughterhouses, we implemented surveillance at 18 and 36 slaughterhouses in which 134 herds from each slaughterhouse were tested, allowing us to compare the effectiveness of expanded surveillance programs with the existing program. Additionally, we ran the programs in which the slaughterhouses were chosen at random (with equal weights across all slaughterhouses) for each simulation run to assess the benefits of surveying only the largest slaughterhouses.

**Network-based surveillance** Since the pig direct transport network has been shown to play a role in pathogen dissemination, Guinat et al. [87] and Bastard et al. [88] explored targeted surveillance programs to reduce outbreak size and found positive results. To be able to relate our results to these findings, we incorporated network-based surveillance programs in our model to compare across surveillance schemes. For each contact type (direct transport, direct truck sharing, indirect truck sharing, exterior truck, geographic), we created an annual static network. We ranked each farm in the network by degree (number of incoming and outgoing contacts), then chose the 250 top-ranked farms. These farms were then surveyed every 90 days, randomly choosing a start date within the first 90 days of the simulation run. The 90 days were chosen to balance between a realistic and useful approach for farm-based surveillance, adapted to the visiting schedule of pig health services in Switzerland. All pigs from the farms were tested and infectious symptomatic and asymptomatic pigs were identified as positive cases. In addition to the 250 targeted farms selected by their degree in the networks, we also chose 250 farms randomly for each simulation. In this way, we could compare the targeted surveillance with non-targeted surveillance of 250 farms.

**Control: isolation and quarantine** Within the surveillance programs, when an infected herd was inspected, all infected pigs were isolated and all susceptible or exposed pigs were quarantined, preventing them from transmitting the pathogen to any other herd.

## Seasonality and annual trends

Sterchi et al. [32] described the seasonality of the transportation data and noticed a trend of decreasing transports beginning from 2014 to 2017. S2 Fig shows the 7-day average of the number of direct pig transports for 2014, 2016, and 2019. The plot indicates the decline in transports; there was a 10% reduction in direct transports from 2014 to 2019. The seasonal changes are shown; the maximum weekly average of transports for 2019 occurred in mid-June with 556 while the minimum was in later December with 245 transports, a more than 50% reduction. To determine whether disease transmission is affected by seasonality and annual trends of transports, we considered an introduction at four different points in time: winter 2014, spring 2014, winter 2019, and spring 2019.

## Sensitivity analysis of tour networks

To understand the effect of the lack of tour data coverage and the uncertainty of our tour parameters, we performed a sensitivity analysis on the tour records and the tour parameters.

For the tour parameter sensitivity analysis, we increased the three contact parameter rates ($\phi, \psi, \eta$) by factors of 4, 10, 50, 100 and 200. We simulated 1,000 runs of the model without surveillance for each parameter and each factor increase. With the results, we compared the cumulative number of infected farms and the proportion of new cases per transmission pathway.

Because we do not have all tour information for the transports of pigs, we wanted to know by how much we are underestimating the transmission events caused by tour contacts. If tours are responsible for disease dissemination, then an increase in tour contacts would increase the spread of disease. We removed 5% and 10% of the tours and increased the tour parameters by a factor of 200, a necessity in order to observe any between-farm dynamics. If we consider the cases related to tour contacts with a smaller tour data set and compare with the complete tour data set we have, then we could assess how much the total cases change with an increase of tour data. The tour removal was chosen at random for each simulation run. As with the parameter sensitivity, we calculated the cumulative number of infected farms and the proportion of new cases per transmission pathway.

### Sensitivity analysis of index case

We assessed the sensitivity of the index case algorithm. For every additional increase of risk of farm index case potential, we increased the weight by 1, 2 (model default) and 3.

### Metrics for comparison

To establish the importance of each mode of transmission, we calculated the proportion of new cases from each of the six transmission pathways (the direct transfer, the direct truck share, the indirect truck share, the truck driver and exterior truck fomites, geographic, and within farm). We defined new case here as either a newly infected pig (either on a farm or during a transport) or the direct transport of an infected pig to a new farm.

To determine which surveillance scenario performed the best at identifying an outbreak, we compared the earliest date at which an infected pig was detected.

In addition, we calculated the cumulative number of infected farms to compare the differences in disease progression over time and the differences in disease introduction dates.

### Implementation

Simulations were run on UBELIX (http://www.id.unibe.ch/hpc), the HPC cluster at the University of Bern. The model was written in Python version 3.10. The model is available on GitHub at https://github.com/kmoriarty123/SwineNet-model. The data preprocessing and plotting were written in R version 4.0.2.

For all model scenarios, including the various surveillance programs and sensitivity analyses, we simulated 1,000 iterations for 8 months duration. We kept the same seeds across the scenarios in order to ensure the same initial spread dynamics.

## Results

We first ran the model with our base parameters and no surveillance. Fig 5 (A) shows the results of the median, 95$^{th}$ percentile and maximum cumulative infected farms over the first 8 months with a disease introduction in May 2019. On the last day of the simulation, the median number of cumulative infected farms was 1 for PRRS, 112.5 for ASF and 1 APP. Since our objectives were to understand the effect of different contact types on disease dissemination and the effectiveness of the surveillance programs, we focused on the simulations in which the disease spreads, which we defined as causing a large outbreak of 10 or more infected farms when no control measures were implemented. Across all simulations, only 16% of PRRS runs and 11% of APP runs led to large outbreaks, compared to half of the simulations for ASF. The proportion of simulation runs that led to large outbreaks is shown in S3 Fig. Fig 5 (B) differs from (A) in that it only includes the simulations with large outbreaks. On the last day of these simulations, the median number of cumulative infected farms was 39 for PRRS, 6,333 for ASF and 22.5 for APP. ASF propagates rapidly, while APP and PRRS have slower rates of progression. A similar plot of median, 95$^{th}$ percentile and maximum cumulative number of infected pigs can be found in S4 Fig.

**Seasonality and annual trends** The changes in median cumulative infected farms of the different seasonal introduction dates can be seen in Fig 6. For PRRS, the largest difference between the median cumulative sum of infected farms after 242 days of model run was 6 (an increase of 19%), while it was 8 (42%) for APP. For ASF, a difference between the median values materializes after day 125 and a median difference of 1,527 (24%) on day 242. We used the Kolmogorov–Smirnov two-sample test to determine if the ASF median cumulative infected number of farms between the seasons (2014-01-01, 2014-05-01) (2019-01-01, 2019-05-01)

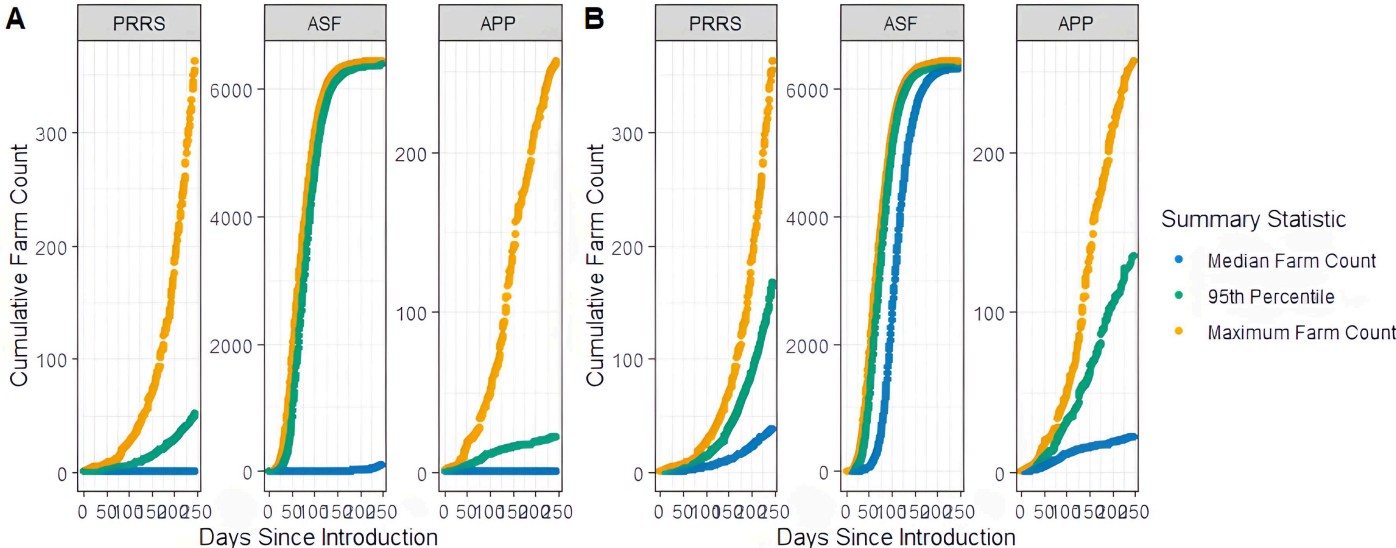

**Fig 5. Median, 95th percentile, and maximum cumulative infected number of farms for each disease without incorporating surveillance and a date of disease introduction in May, 2019.** (A) All 1,000 simulation runs were included. (B) Only simulation runs with large outbreaks, as defined by 10 or more cumulative infected farms.

and the years (2014-01-01, 2019-01-01) (2014-05-01, 2019-05-01) were from different cumulative distributions. We could not reject the null hypothesis (unadjusted p-values of 0.96, 0.96, 0.32, 0.45), indicating no evidence that the difference in introduction date results in significantly different disease trajectories for ASF. A plot of the cumulative probability curves can be seen in S5 Fig.

**Routes of transmission** The results of our comparison of the relative impact of the six transmission pathways can be seen in Fig 7 (A).

Throughout the model run, more than 76% of new cases were due to within-farm spread. The remaining cases were mostly through direct transport of pigs and geographic spread. Fig 7 (B) presents the same comparison as Fig 7 (A), but it excludes the within-farm spread cases in order to compare the transmission dynamics of only the between-farm spread. Among the between-farm pathways, tour contacts accounted for less than 1% of new cases, regardless of disease. However, it is important to note that the tour contact data were incomplete and may underestimate the true role of this pathway. To address this uncertainty, we conducted sensitivity analyses that scaled up tour-related parameters. These analyses showed that even with significant multiplication of tour frequency and associated risk parameters, the overall contribution of tour transmission remained low. For APP, only the direct transfer of pigs is important through the 8-month simulation run. For ASF, geographic transmission is the most prominent for the first 118 days of an outbreak, but then direct truck transfer becomes more important. Similarly for PRRS, direct truck transfer and geographic spread are drivers of disease dissemination, but only nearly 50 days after disease introduction.

**Surveillance** For each simulation run where the scenario without surveillance led to large outbreaks, we determined the date the first case was detected (Fig 8).

The most effective surveillance programs to detect an outbreak of PRRS were farmer-based with a 20% morbidity threshold and with 90% and 100% of farmer participation. This occurred, on average, after 93 days of disease introduction, at which point 5 farms and 36 pigs were infected. Ten days later, on day 102, the surveillance programs of 40% morbidity with

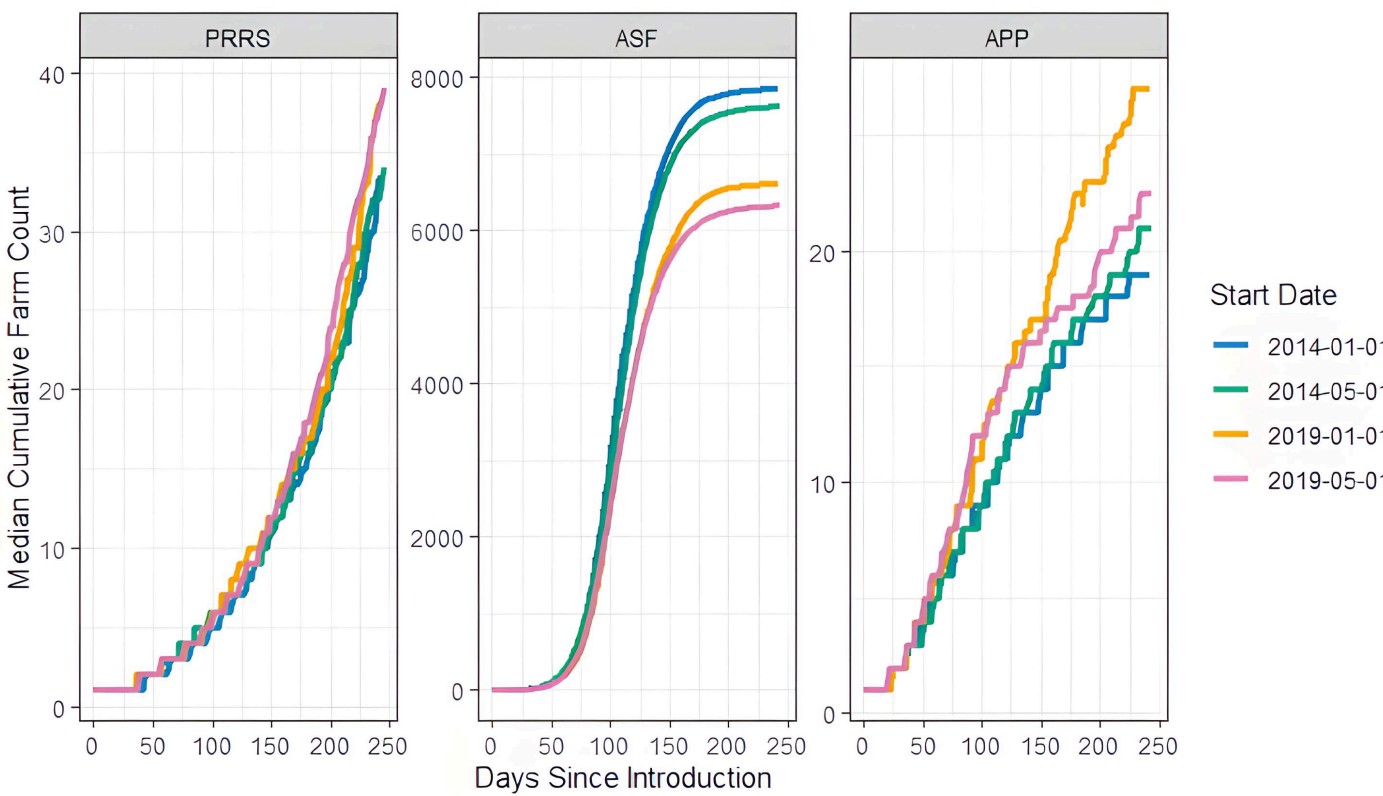

**Fig 6. Seasonal changes.** Median cumulative infected farms by date of pathogen introduction for simulations with large outbreaks.

the same farmer participation level detected the first cases. One week after that, on day 108, the farmer-based with 60% of farmer participation detected the first cases when 6 farms and 53 pigs were infected. Nearly three months later, the farmer-based mortality programs began detecting cases. The first of the network-based surveillance, the indirect truck share network, detected sick pigs at 229 days. At this point, 33 farms were infected with a total of almost 500 sick pigs. Finally, none of the slaughter-based surveillance programs could detect cases.

As with PRRS, the farmer-based surveillance performed the best in detecting an outbreak of ASF. Within 31 days, on average, all farmer-based surveillance programs were able to detect the first case of ASF regardless of farmer participation or morbidity and mortality threshold. The first of these was at 23 days; the surveillance programs with 90% and 100% farmer participation and 20% morbidity began detecting cases. Three days later, on day 26, the farmer-based with 60% of farmer participation detected a case with 20% morbidity. The surveillance at 9, 18 and 36 slaughterhouses resulted in, on average, an outbreak detection after 113, 69, and 55 days, respectively. When the 36 slaughterhouse program detected the first case at 55 days, 131 farms and more than 11,000 pigs had been infected with ASF. When the 9 slaughterhouse program detected the first case at 113 days, 3,625 farms had been infected with more than 660,000 sick pigs. The 36 slaughterhouse-based program detected the first case almost two months before the 9 slaughterhouse-based program, a difference of more than 650,000 sick pigs. Of the network-based surveillance, the direct truck share network and the exterior truck fomite network detected the first case, on average, at 82 days. At this point,

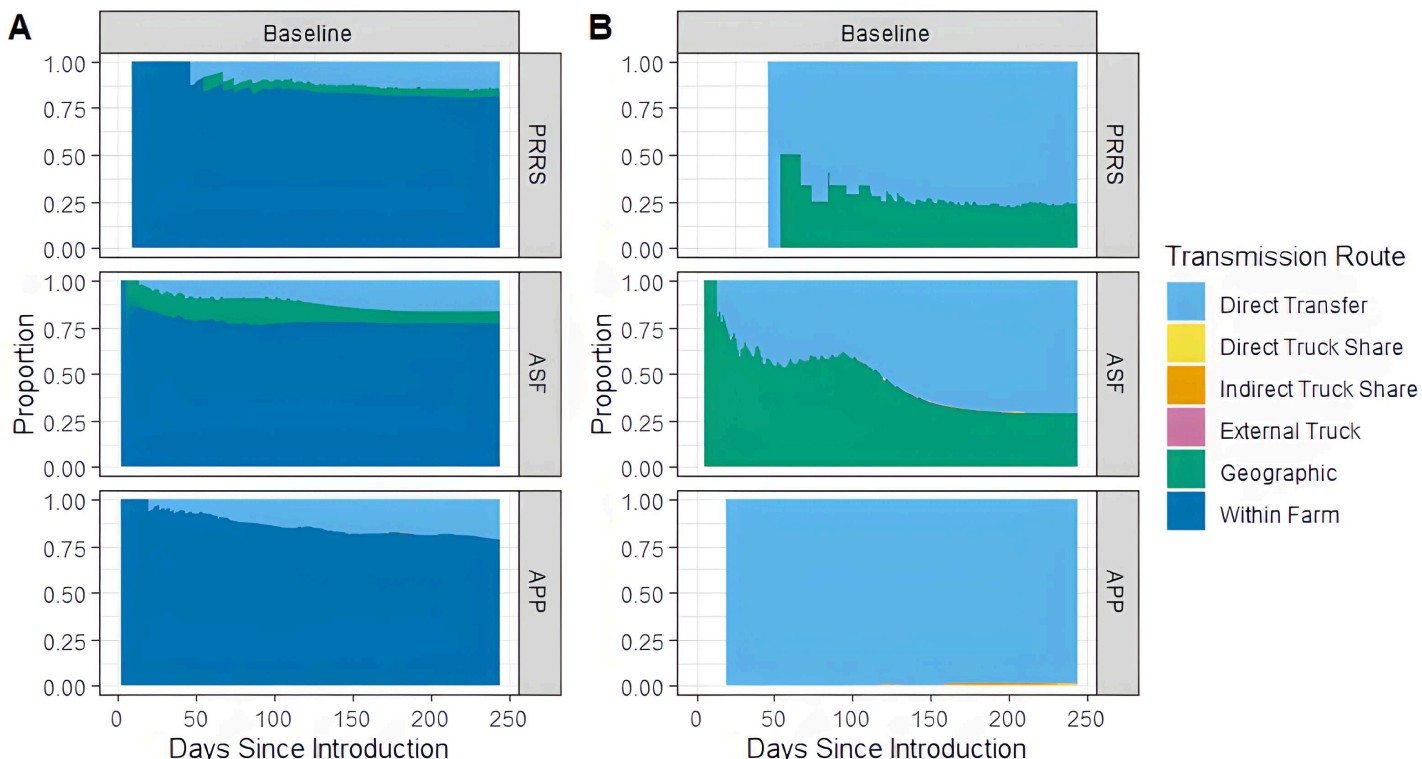

**Fig 7. Proportion of new cases transmitted via each transmission pathway for only simulations with large outbreaks and a disease introduction of May 2019.**
(A) Baseline parameters including within farm spread (B) Baseline parameters excluding within-farm spread.

over 100,000 pigs were already infected. This was only 4 days earlier than if 250 farms were randomly selected.

For APP, similarly to PRRS and ASF, farmer-based surveillance programs were the first to detect cases of APP. With 90% and 100% farmer participation and 20% morbidity, the surveillance system detected the first case at 45 days. It took 7 more days for 60% farmer participation, at which point 5 more farms were infected and 330 pigs, on average. All farmer-based programs based on morbidity found the first case within 63 days. The exterior truck fomite network-based surveillance programs detected the first case 153 days after introduction, when almost 2,800 pigs were infected from 17 farms. The surveillance programs based on direct transfer networks and the direct truck share networks resulted in detection weeks later at 170 and 176 days. All other surveillance programs were, on average, not able to detect a case over the 8-month simulation run.

The slaughterhouse-based surveillance was implemented, targeting the 9, 18 and 36 largest slaughterhouses. We additionally ran the model to simulate scenarios where the 9, 18 and 36 slaughterhouses were chosen randomly for each simulation run. We found that choosing the largest slaughterhouses resulted in an earlier detection of pigs for 18 and 36 slaughterhouse programs, but when surveying only 9 slaughterhouses, choosing the largest or random does not have an impact on the earliest date of detection. Choosing the 18 and 36 largest slaughterhouses resulted in a first detection day of 69 and 55, respectively. For both schemes, it was 28 days earlier than if the slaughterhouses were chosen randomly. These results are shown in S6 Fig.

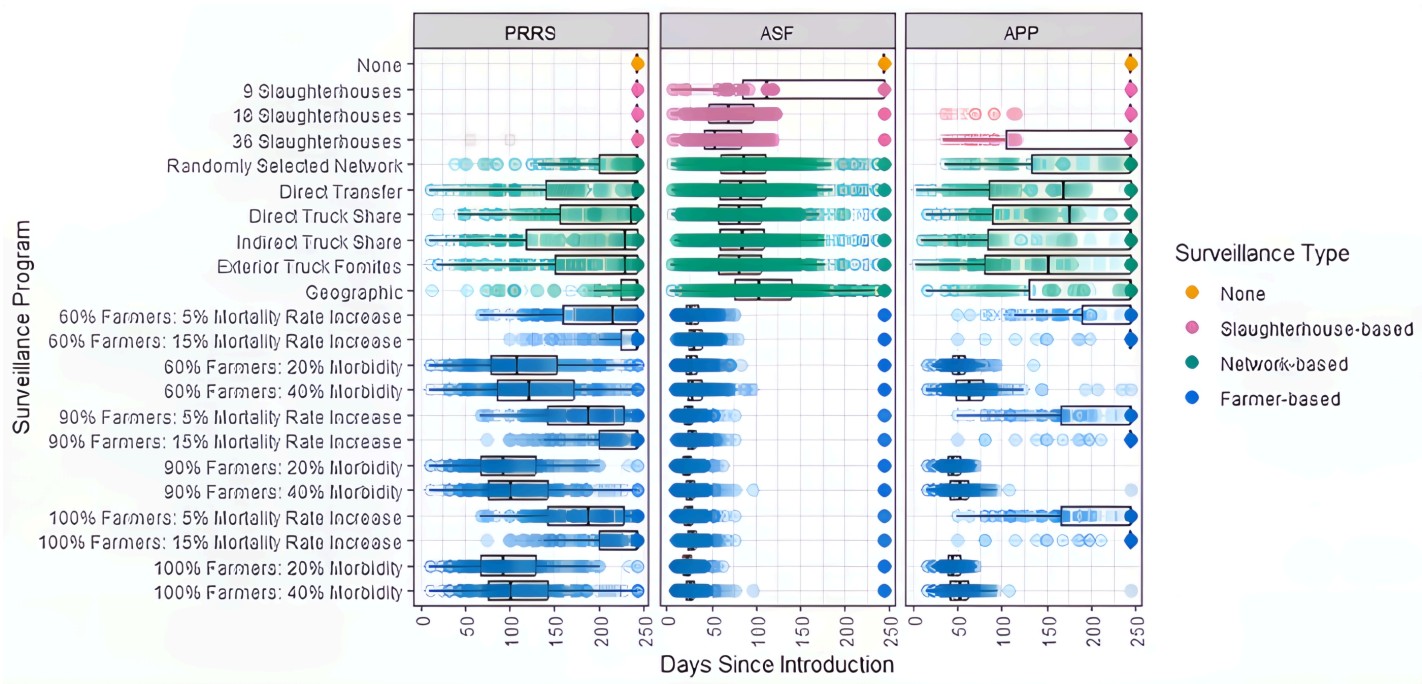

**Fig 8. Earliest date at which an infected pig was detected.** Each circle represents the first date a positive case was detected for the simulation run. The boxplot shows the distribution of these dates, with the box representing the interquartile range (Q1 to Q3) and the line indicating the median. The only simulations included are those that lead to large outbreaks at the end of the 8 months simulation period in the scenario excluding control measures or surveillance. The disease introduction is May 2019.

**Sensitivity analysis of tour parameters $\phi$, $\psi$ and $\eta$ and tour data** From the sensitivity analysis of our tour parameters of the direct truck share ($\phi$), indirect truck share ($\psi$), and exterior truck fomite ($\eta$), results of the parameter increase of 50, 100, and 200 are shown in S7 Fig. In the first 200 days of disease spread, the maximum difference is less than 1% in the cumulative number of infected farms. For APP on day 244, there is a difference of 4 infected farms between the baseline indirect truck share parameter and the corresponding parameter increased by a factor of 200. For PRRS, there was a difference of 9 infected farms between the model with baseline exterior truck tires parameter and the parameter increased by a factor of 200.

In addition, we calculated the percentages of cases from each of the contact transmission pathways out of all new cases after the increase in tour parameters by factors of 50 and 100 (see S8 Fig (B)–(D)). Similar to the findings of the cumulative number of infected farms, the maximum change in these percentages was less than 3% except for the indirect truck share parameter of APP beginning after day 100. On day 244, the percentage of new APP cases due to indirect truck share increased from 0% (baseline $\phi_{APP}$) to 8% ($100 \times \phi_{APP}$). When $\phi_{APP}$ increased by a factor of 10, the increase in percentage of cases was less than 2%. Results from the APP indirect truck share parameter increase of 4, 10, 50 and 100 is shown in S9 Fig.

When we removed the tour data randomly by 5% and 10% and increased the tour disease transmission parameters by a factor of 200, we found there was no statistically significant difference (Mann Whitney U-test p-value = 0.33 for PRRS) in the median cumulative number of infected farms when the model was run using a 200 factor increase in tour parameters

with 10% tour reduction and no tour reduction. Results are shown in S10 Fig. More detailed information can be found in S1 Appendix Section A and Section B.

We also calculated the proportion of new cases per transmission pathway with the tour parameters increased by a factor of 200 and the baseline, 5% and 10% tour reductions. We found no changes in the transmission dynamics. The results are shown in S11 Fig.

**Sensitivity analysis of index case** We assessed the sensitivity of the index case selection algorithm. We found that when the index case weighting scheme factor decreases to 1 or increases to 3, the cumulative number of infected farms and cumulative number of infected pigs are not affected. Results can be found in S12 Fig along with detailed information in S1 Appendix Section A and Section B.

## Discussion

We developed a model that simulated the spread of multiple pathogens, independently, along the contact networks of the Swiss swine sector. Our model simulation results indicated that ASF has the potential to cause a substantial number of swine deaths if no surveillance programs were implemented. The alarming results of large numbers of ASF-positive cases are consistent with what was witnessed across Asia and several European countries [1,89]. Our model enables us to compare the several surveillance programs we implemented and to provide insight into the relative importance of distinct transmission pathways.

The farmer-based surveillance programs, in which farmers observe their animals daily, detected outbreaks the earliest, even with only 60% of farmer participation, regardless of the pathogen modeled in this study. The best performing surveillance scenario would be 100% farmer participation at 20% morbidity. However, there was little difference between 90% and 100% farmer participation while, surprisingly, 60% participation was only slightly worse (PRRS: 24 days later, ASF: 3 days later, APP: 6 days later). For ASF, all farmer-based surveillance detected the first case within the first month of the outbreak. For APP and PRRS, the farmer-based morbidity programs were much more effective than the mortality-based programs, due to their lower mortality rates. Similar to our positive findings of farmer-based surveillance, Garner et al. [90] used a simulation model to compare surveillance strategies for Foot and Mouth Disease in Australia. They compared the current passive surveillance program of farmer reporting with active surveillance of testing pigs at saleyards and bulk milk testing of dairy cows. They found that between the three programs compared, the one program that detected disease the earliest was farmer surveillance when 10% of the herd showed clinical signs. In our study, we initiated farmer-based surveillance at 20, 30, and 40% of the herd with clinical signs, and we also concluded that these were the most effective strategies.

With the exception of a few simulation runs, the slaughter-based surveillance was not able to identify an outbreak of PRRS or APP within the 244 days of the simulated outbreaks. When the slaughter-based surveillance first detected ASF, more than 10,000 pigs would have been infected with the virus. Thus, our modeling analysis suggests that the slaughter-based surveillance is not sufficient for early detection of outbreaks of PRRS, ASF, or APP. However, a program surveying the 36 largest slaughterhouses is more effective in detecting ASF compared to a program focused on only the 9 largest slaughterhouses, identifying an average of over 3,500 additional infected farms. In these surveillance programs, the herds are surveyed daily until 134 herds from each of the slaughterhouses have been investigated. If more slaughterhouses are enlisted and more than 134 farms are tested at each slaughterhouse, then the slaughterhouse surveillance might be effective for pathogens comparable to ASF. Correia et al. [91] evaluated targeted surveillance programs of slaughtered pigs using data from

swine in Great Britain. For diseases with low prevalence, they suggest these slaughterhouse-based programs are inefficient at identifying outbreaks, as was the case in our model. In the case of CSF, Crauwels et al. [92] found that routine serological exams have a low likelihood of detecting outbreaks. The routine serological exams apply not only to the implemented slaughterhouse-based surveillance, but also the targeted network-based surveillance.

The network-based surveillance, with frequency of testing every 90 days, first detected outbreaks after many pigs were already infected — 500 for PRRS, 100,000 for ASF and 2,800 for APP. Guinat et al. [87] ran network analyses on pig movements in the Russian Federation and concluded that targeted surveillance of farms based on static network metrics of degree may result in better detection of outbreaks, reducing the size of the outbreak. Their findings suggested that increasing surveillance at pig holdings that were highly connected with other farms through transport and distance, resulted in faster detection of disease outbreaks. In order to test this hypothesis, many network simulation models have been created. Bastard et al. [88] found that targeted surveillance based on the in-degree (the number of incoming contacts) and out-degree (the number of outgoing contacts) metrics of a pig holding optimized the targeted surveillance of livestock-associated methicillin resistant *Staphylococcus aureus*.

The results from our model show that the targeted surveillance of 250 farms selected based on their degree performed better than 250 randomly selected farms only in a few scenarios. This is contrary to what Bastard et al. [88] found. Bastard et al. [88] had surveyed 30, 60, and 120 farms. The frequency of the surveillance could be continuous, in which case the structure of their program is less realistic than our every 90 day surveillance. The direct transfer, the direct-truck share and the truck fomite network-based targeted surveillance performed better at detecting an outbreak of APP two to three months earlier than the surveillance of randomly selected farms. The results of our simulations indicate that targeted surveillance based on network degree centrality might be a potential surveillance program for pathogens similar to APP if more farms are chosen and surveyed more frequently.

Our model outcomes suggest that an investment in farmer-based surveillance programs could be the most effective approach for early detection. The testing efforts in the most effective slaughterhouse-based surveillance program, the 36 largest slaughterhouses, resulted in approximately 30,550 tests over 8 months (36 slaughterhouses, 134 batches per slaughterhouse, 6 pigs per batch). Therefore, our model suggests that the number of tests used in effective slaughterhouse programs would need to be increased to more than 30,550 by either selecting more slaughterhouses to survey, increasing the number of pigs randomly selected or changing the frequency of testing. For the network-based surveillance, more than 250 farms would need to be visited more than every 90 days in order to detect outbreaks early. This would require over 1,000 farm visits and more than 393,000 tests per year (250 farms, an average of 393 pigs on the targeted-farms, 4 times per year). However, a careful cost-benefit analysis is needed to clarify the investment required for sufficient uptake of farmer-based surveillance. Such analysis would allow for a comparison with the testing efforts in slaughter- and network-based programs along with combinations of these programs.

The recent report by the European Food Safety Authority (EFSA) [61] which describes the current epidemiology of ASF, affecting 14 member states in 2023, found that 94% of ASF outbreaks among domestic pigs were detected through passive surveillance based on clinical suspicion. This aligns closely with our study, which identified farmer-based surveillance as the most effective approach for early outbreak detection. This parallel supports our conclusion that proactive farmer observation plays a critical role in identifying disease outbreaks sooner, even at lower levels of participation. EFSA's epidemiological analysis reinforces the need to

implement robust farmer-centered surveillance programs to mitigate the rapid spread of ASF and other swine diseases, as illustrated by our model.

Our assumption of homogeneous mixing within herds simplifies the model by assuming all pigs have equal chances of interacting and transmitting disease, potentially overlooking more complex within-farm dynamics influenced by factors such as herd type and housing system. Additionally, our use of a frequency-dependent transmission model, which assumes higher per-pig transmission rates on smaller farms, could oversimplify dynamics on larger farms. Considering that on small farms, pigs are typically housed in single bays, whereas on larger farms they are kept in several bays or even separate buildings, higher contact rates between individuals are expected on small farms. Nevertheless, the assumptions applied in our model were suitable for studying between-herd transmission. A more detailed within-farm model would require better data on transmission, asymptomatic, and recovery rates of individual pigs as well as specific farm-level biosecurity measures, which was beyond the scope of our study.

Due to the limited published estimates on transport tour transmission rates of direct and indirect truck sharing and truck fomites, setting the parameter values for our model was difficult. Galli et al. [93] interviewed several Swiss pig health and logistics experts who identified likely transmission pathways for PRRS and ASF. Based on their expertise, they concluded that PRRS had a higher or equal likelihood of transmission than ASF for all tour pathways. Our model parameters are not consistent with these conclusions because we based the tour parameters for our model on the direct transmission rates found in countries where ASF and PRRS are persistent. Accordingly, significant uncertainty remains on the values of these parameters. As a sensitivity test, we increased the tour parameters by a factor of 100, and we found there was little change in the cumulative number of infected farms. This suggests that even if these parameters are not correct, they would need to be inaccurate by a factor of more than 100 in order to have an effect on the cumulative number of infected farms. In contrast, after increasing the indirect truck share parameter by a factor of 100 for APP ($100 \times \psi_{APP} = 1$), the proportion of cases due to indirect truck share increased from less than 1% to 8%. Previous experimental studies suggest that $\psi_{APP}$ is much lower than 1 [50,81]. When the parameter increased by a factor of 4 or 10, resulting in more realistic values of $\psi_{APP}$ (0.04, 0.1, respectively), the proportion of cases due to indirect truck share increased but remained under 2%.

Besides effective surveillance identification, the other objective of our model was to identify routes of disease transmission. Several studies have concluded that there is a potential risk for disease spread along tour networks [8,94,95]. Porphyre et al. [94] created a static network of the British swine tour network. Using descriptive network analysis, they concluded that tour networks have a significant potential for spreading disease. From our model outcome of 8 months-long simulations, contact via tours and exterior truck fomites were, on average, the source of less than 1% of new cases. In contast, within-farm spread caused most cases and both geographic and direct transfer of pigs caused additional dispersion. The percentage of new cases due to between-farm contact was PRRS: 20%, ASF: 24%, and APP: 22%. Of these between-farm cases, less than 1% was due to tour contacts. Even with sensitivity analysis of the tour parameters and tour reduction, we found no evidence that tour vehicles and drivers were responsible for dispersion of the pathogens in comparison to direct transport and local modes of transmission. A recent network-based model of PRRS in the United States found that 15–20% of the between-farm cases were due to contact via the transportation truck, either by indirect truck sharing or truck fomites [96]. These results suggest that complete tour data could impact the simulated model results for PRRS.

Even with sensitivity analysis of the tour parameters and tour reduction, we found little evidence that tour vehicles and drivers were responsible for pathogen dispersion compared to

direct transport and local modes of transmission, though the limited representation of tour data (only 15% of the TVD transports) means these results should be interpreted with caution, as we are unable to evaluate whether the remaining 85% plays a disproportionately larger role in transmission.

Finally, the seasonal differences found in the transport network did not reflect changes in the trajectories of disease dissemination as network analysis studies have suggested in Sweden or the Russian Federation [87,97]. Our analysis suggests that seasonality plays less of a role in the Swiss setting.

Unfortunately, we could not validate our model to disease data since positive cases of PRRS and APP are very rare in Switzerland, and ASF has yet to reach it. These few cases are reflected in our model predictions with only 16% of PRRS and 11% of APP simulations resulting in outbreaks. The illustrative aspect of our model allows us to compare disease trajectory, surveillance program effectiveness, and transmission pathway proportions, but not the breadth and duration of disease outbreaks. We would need more disease data to test the robustness of the model parameters before drawing generalizable recommendations from these disease-specific outbreak metrics.

In order to limit the number of estimated parameters, we constructed a relatively simple model. Expanding the model to capture complex disease dynamics would require additional data for appropriate model calibration. With simplicity comes limitations in capturing disease idiosyncrasies, such as herd-type or age-dependence of pig diseases. With the exception of PRRS, we assumed that the same within-herd and between-herd dynamics were applied to all herds. APP can affect pigs of all ages, but older, fattening pigs, might have higher acquired immunity, which may result in different disease dynamics than those captured here [98].

On the other hand, the detailed data on transport of pigs within Switzerland allowed us to focus our model on capturing the key aspects of disease spread. The contact networks we developed in conjunction with the surveillance systems we simulated support decision makers looking to compare the different surveillance measures before a new pathogen is introduced.

The simulation results revealed that ASF outbreaks were significantly larger in scale compared to PRRS and APP, highlighting the substantial threat ASF poses. To better prepare for its introduction into the Swiss swine industry, further research is needed. Expanding the model to capture the spread of ASF among wild boar, farm-specific population dynamics, nuanced seasonal changes with varying dates of introduction, the effect of additional tour data, and the outcome of ASF-specific control measures would provide further insight into the introduction and spread of ASF among Swiss swine. These improvements could refine the accuracy of model trajectories by incorporating more detailed information on disease reservoirs, transmission patterns, and the effectiveness of control measures.

## Supporting information

**S1 Tables. Supplementary tables.** Table A: Pig holding and tour characteristics; Table B: PRRS index case weights; and Table C: APP and ASF index case weights.
(PDF)

**S1 Fig. The distribution of pig holdings by their annual mean number of pigs per year for pig holdings with less than 2,000 pigs.**
(TIFF)

**S2 Fig. The daily average of pig transports over the previous seven days for 2014, 2016, and 2019.** January 1, 2014 through January 6, 2014 are not included.
(TIFF)

**S3 Fig. Proportion of simulations by introduction date that had a large outbreak by disease.** A simulation is classified as having a large outbreak when at least 10 or more farms have been infected.
(TIFF)

**S4 Fig. Median, 95th percentile, and maximum cumulative infected number of pigs for each disease without incorporating surveillance and a date of disease introduction in May, 2019.** (A) All 1,000 simulation runs were included. (B) Only simulation runs with large outbreaks, as defined by 10 or more cumulative infected farms.
(TIFF)

**S5 Fig. Cumulative probability of median infected number of farms for ASF for each date of introduction.**
(TIFF)

**S6 Fig. Earliest date at which an infected pig was detected for surveillance programs with largest selected and randomly selected slaughterhouses.** Each circle represents the first date a positive case was detected for the simulation run. Only simulations that lead to large outbreaks at the end of the 8 months simulation period considering no control measures are included.
(TIFF)

**S7 Fig. Median cumulative infected number of farms by adjustments to factors of $\phi$, $\psi$ and $\eta$.** $\phi$ is the direct truck share transmission parameter, $\psi$ is indirect truck share transmission parameter, and $\eta$ is exterior truck and truck driver transmission parameter. Results are from a disease introduction from May, 2019 and from simulations with large outbreaks only. We plot only the baseline and factors beginning at 50 because smaller values resulted in minimal change in cumulative number of infected farms.
(TIFF)

**S8 Fig. Proportion of new cases transmitted via each route for only simulations with large outbreaks and a disease introduction of May 2019.** (A) Baseline parameters (B) An increase of the direct truck share parameter ($\phi$) by a factor of 50 (+ Direct) and 100 (++ Direct) (C) An increase of indirect truck share parameter ($\psi$) by a factor of 50 (+ Indirect) and 100 (++ Indirect) (D) An increase of the exterior truck fomite parameter by a factor of 50 (+ Exterior) and 100 (++ Exterior).
(TIFF)

**S9 Fig. Proportion of new cases transmitted via each route after an increase of indirect truck share parameter ($\psi$) by a factor of 4 ($\times$4 Indirect), 10 ($\times$10 Indirect), 50 ($\times$50 Indirect), and 100 ($\times$100 Indirect) for only simulations with large outbreaks and an APP introduction of May 2019.**
(TIFF)

**S10 Fig. First quartile, median, and third quartile cumulative number of infected farms by proportion of removed tours.** Only simulations with large outbreaks and with a disease introduction in May 2019 are included.
(TIFF)

**S11 Fig. Proportion of new cases transmitted via each transmission pathway while removing a percentage of tours.** All Tours indicates the results with all of our current tour data, -5% are results after a random selection of 5% of tours are removed, -10% is the same but for 10%.

(+++) indicates that the tour parameters are increased by a factor of 200. The model results are only simulations with large outbreaks and with a disease introduction in May 2019. (TIFF)

**S12 Fig. Index case weight sensitivity analysis.** Median cumulative number of infected farms (A) and pigs (B) for each disease with different index weight schemes. Index case weight factor 1, is the model run with uniform random selection of farms while factor 3 is the model run where each increase level of risk for the farm, increases the weight of index case selection by a factor of 3. No surveillance and an introduction date in May 2019 were used. (TIFF)

**S1 Appendix. Description of sensitivity analyses on tour data and index case weighting.** (PDF)

## Acknowledgments

Calculations were performed on UBELIX (http://www.id.unibe.ch/hpc), the HPC cluster at the University of Bern. The data was provided by the Identitas AG from the Federal Office of Agriculture (FOAG) and private pig traders.

## Author contributions

**Conceptualization:** Kathleen Moriarty, Antoine Champetier, Salome Dürr, Nakul Chitnis.

**Data curation:** Kathleen Moriarty, Francesco Galli.

**Formal analysis:** Kathleen Moriarty.

**Funding acquisition:** Salome Dürr.

**Methodology:** Kathleen Moriarty, Antoine Champetier, Nakul Chitnis.

**Supervision:** Salome Dürr, Nakul Chitnis.

**Visualization:** Kathleen Moriarty.

**Writing – original draft:** Kathleen Moriarty, Nakul Chitnis.

**Writing – review & editing:** Kathleen Moriarty, Antoine Champetier, Francesco Galli, Salome Dürr, Nakul Chitnis.

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
