## [Decision Letter · Decision Letter 0]

18 Mar 2025

PONE-D-25-03280Identifying effective surveillance measures for swine pathogens using contact networks and mathematical  modelingPLOS ONE

Dear Dr. Moriarty,

Thank you for submitting your manuscript to PLOS ONE. After careful consideration, we feel that it has merit but does not fully meet PLOS ONE’s publication criteria as it currently stands. Therefore, we invite you to submit a revised version of the manuscript that addresses the points raised during the review process.

**ACADEMIC EDITOR:**
Please below are comments, including attached files.

We look forward to receiving your revised manuscript.

Kind regards,

Charles Odilichukwu R. Okpala, PhD

Academic Editor

PLOS ONE

Journal Requirements:

3. Thank you for stating the following in the Acknowledgments Section of your manuscript: This project was partially funded by the Swiss National Science Foundation (SNSF; project number 182404). Calculations were

performed on UBELIX (http://www.id.unibe.ch/hpc), the HPC cluster at the University of Bern. The data was provided by the

Identitas AG from the Federal Office of Agriculture (FOAG) and private pig traders.

 SD, AC, and FG were supported by the Swiss National Science Foundation (SNSF:

https://www.snf.ch/en; project number 182404). The funders had no role in study

design, data collection and analysis, decision to publish, or preparation of the

manuscript.

Additional Editor Comments:

Please attend to the comments of the reviewers.

Reviewers' comments:

Reviewer's Responses to Questions

**Comments to the Author**

1. Is the manuscript technically sound, and do the data support the conclusions?

Reviewer #1: Yes

Reviewer #2: Yes

Reviewer #3: Yes

2. Has the statistical analysis been performed appropriately and rigorously? 

Reviewer #1: Yes

Reviewer #2: Yes

Reviewer #3: Yes

3. Have the authors made all data underlying the findings in their manuscript fully available?

Reviewer #1: Yes

Reviewer #2: Yes

Reviewer #3: Yes

4. Is the manuscript presented in an intelligible fashion and written in standard English?

Reviewer #1: Yes

Reviewer #2: Yes

Reviewer #3: Yes

5. Review Comments to the Author

Reviewer #1: The manuscript presents a critical analysis of how swine pathogens disseminate through multiple transmission pathways, including direct transport, truck tours, and local spread, while also assessing the effectiveness of various surveillance strategies in detecting these pathogens in a timely manner. The study is well-structured, addressing an important issue in animal health and epidemiology, with potential implications for biosecurity and disease control measures.

Reviewer #2: Reviewer’s Comments:

Manuscript ID: PONE-D-25-03280

Title: Identifying effective surveillance measures for swine pathogens using contact networks and mathematical modeling

The Academic Editor,

POLS One.

Dear Dr. Charles Odilichukwu R. Okpala, PhD,

Thank you for recognizing me to review this manuscript. Generally speaking, this work is super excellent. I found the article so interesting and insightful, where all the sections, from the introduction to the conclusion, were well written and rich in content. It conveyed scientific information about the use of contact networks and mathematical modeling techniques in the identification of effective surveillance measures for swine pathogens. The authors have done a great job of reporting their findings to the scientific community regarding the effectiveness of surveillance measures in identifying the specific pathways of disease spread from one place to another. However, there are a few questions and suggestions I found attractive, and if properly addressed or cleared the doubt of the readers, then that would improve the work and reduce the ambiguity therein.

(A) Changes need to be made throughout the entire manuscript

i. The authors should be consistent in using either American or British type of spelling throughout the entire text, e.g., the spelling of a word modelling/modeling.

Reviewer #3: The study is well-structured, methodologically rigorous, and addresses an important topic in veterinary epidemiology. The use of empirical Swiss transport data and stochastic modeling provides valuable insights into disease spread dynamics. The findings on farmer-based surveillance as the most effective strategy align with real-world observations (e.g., EFSA reports) and add practical relevance.

Key Recommendations for Revision

1. Data availability

The current statement explains restrictions due to third-party data (Identitas AG and FOAG). To comply with PLOS ONE’s data policy, ensure the following:

Explicitly state the criteria for accessing data (e.g., contact information for Identitas AG and FOAG).

Confirm that data will be available indefinitely upon request.

2. Figures and Tables

Figure Quality: Ensure high-resolution figures with axis labels, legends, and annotations. For example:

Figure 1: Add a key explaining symbols (PRRS/APP cases, pig holdings).

Figures 5–8: Clarify if medians/percentiles are shown for outbreaks or all simulations.

Figure Citations: Reference figures in the correct order (e.g., Figure 3a/3b descriptions appear before Figure 4 in the text but are listed afterward).

Table 1: Format consistently (e.g., superscripts for citations in the "Routes of Transmission" row).

3. Clarity and language

Simplify complex sentences (e.g., in the Methods: "The stochasticity of the model is generated from the τ-leap method..." → "We used the τ-leap method with Poisson and binomial sampling to model stochasticity...").

Define all acronyms at first mention (e.g., SEIAR in the Abstract).

4. Methodological Limitations

Tour data coverage: Emphasize that only 15% of transport tours were included and discuss how this might underestimate transmission via indirect contacts. Expand the sensitivity analysis discussion to highlight this limitation.

Homogeneous mixing assumption: While acknowledged, briefly suggest how future work could incorporate heterogeneous mixing (e.g., farm size or housing systems).

5. References

Ensure all in-text citations (e.g., "You et al., 2021") match the reference list.

Format references consistently (e.g., journal names in italics, proper use of "et al.").

Minor Issues

Abstract: Specify the time frame for ASF detection ("8 weeks or more before") relative to other surveillance methods.

Results: In Figure 8, clarify whether error bars or confidence intervals are included for detection timelines.

Supplemental Material: Ensure all supplementary figures/tables are cited in the main text (e.g., "Suppl. Figure 3" is mentioned but not clearly linked to a specific result).

6. PLOS authors have the option to publish the peer review history of their article (what does this mean?). If published, this will include your full peer review and any attached files.

Reviewer #1: **Yes: **Mohamed A. Bakheet

Reviewer #2: No

Reviewer #3: No

---

## [Author Response · Author response to Decision Letter 1]

4 May 2025

The response to reviewers is included in the attached document titled "Response to Reviewers." For your convenience, I’ve also copied the contents of that PDF below. Please note that in the original PDF, our responses to the reviewers’ comments are highlighted in red font for clarity.

Reviewer #1

General Comments

1. Relevance and Significance

· The manuscript explores a crucial topic, examining the spread of swine pathogens through multiple transmission pathways including direct transport, truck tours, and local spread and evaluating the effectiveness of different surveillance strategies in timely pathogen detection.

· The comparative approach using PRRS, ASF, and APP which have distinct biological and epidemiological profiles is particularly valuable. It highlights how differences in transmission modes affect the dynamics and detection of disease in pig populations.

2. Novelty

· The integration of three data sources (TVD records, pig trader data, and AGIS holdings data) to construct a complex, multi-layered contact network is a strong point of the study.

· Addressing both endemic (APP, PRRS) and emerging threats (ASF) in one modeling framework is quite innovative and provides a broader epidemiological scope than single-disease models.

3. Methodological Approach

· Overall, the approach (SEIAR within-herd model + dynamic, labeled, between-farm contact network) is logical and seems to capture key processes of disease transmission.

· The differentiation among direct transport, indirect truck-sharing, and fomites via truck drivers/equipment is valuable but still relatively underexplored in the literature. The authors make a genuine effort to incorporate real-world heterogeneities where data allow.

4. Clarity and Organization

· The manuscript is generally well-organized. The distinct sections (Introduction, Methods, Results, Discussion) are clearly delineated.

· Some sections, especially the Results, occasionally become quite dense when describing multiple pathogens and numerous surveillance scenarios. Additional structuring of the results possibly using subsections with descriptive headings would improve readability.

We acknowledge that the Results section covers a range of analyses across the different pathogens and surveillance scenarios, which has led to a high level of detail. To improve the clarity and readability, we had already structured the Results section into 5 subsections: (1) Seasonality and annual trends, (2) Routes of transmission, (3) Surveillance, (4) Sensitivity analysis of tour parameters ϕ, ψ and η and tour data, and (5) Sensitivity analysis of index case. We prefer this to the alternative of organising the Results by disease because it allows us to focus on aspects of transmission and control across diseases rather than focus on particular diseases.

5. Limitations

· The study acknowledges data gaps on truck tours (only 15% coverage), which constrains definitive conclusions on indirect and fomite-based transmission. While the sensitivity analyses mitigate these concerns somewhat, the limitations should be emphasized more.

We agree that the limited availability of the full truck tour data is a limitation in our model. We have emphasized this in the Section 2.2 Sources of Data, Section 2.10 Sensitivity analysis and the Discussion. A more detailed response addressing this point is included in our reply to the relevant Section-by-Section Comment below.

· The homogeneous mixing assumption at the farm level simplifies within-herd dynamics. Further consideration of farm size and housing system might affect the results, especially for highly transmissible pathogens (ASF) versus those with more variable clinical presentations (APP, PRRS).

We acknowledge that the homogeneous mixing assumption can impact the model outcomes. We have expanded our justification for using homogeneous mixing with frequency-dependent transmission, particularly within the Swiss context where farm density is regulated by law, in Section 2.3 Within-Farm Layer. A more detailed explanation is provided in our response to the relevant Section-by-Section Comment below.

6. Impact

· The findings on how quickly ASF can spread underscore the urgency for robust surveillance measures.

· The conclusions that farmer-based surveillance detects outbreaks earlier than other strategies has direct policy relevance particularly because it aligns with recent real-world experiences of ASF spread in Europe.

· The large potential for ASF spread in a naive country emphasizes the need for preparedness, which this model can inform.

Section-by-Section Comments

Abstract

1. The Abstract clearly summarizes the key objectives (assessment of contact types, evaluation of surveillance strategies).

Thank you for this comment.

2. You might emphasize the key quantitative finding: “Direct truck transport and local spread were found to be the main routes …” This is a strong conclusion that guides the reader upfront.

We have revised this statement to use active voice and added further emphasis. Updated text: Our findings highlight that direct truck transport and local spread are the main routes of between-farm transmission.

3. Consider briefly mentioning the major takeaway on surveillance (farmer-based strategies were the most effective).

The major takeaway regarding surveillance, farmer-based strategies being the most effective, is already stated in the abstract, where we highlight the quantitative differences. Specifically, farmer-based surveillance programs were the only measures that consistently identified outbreaks of APP and PRRS, and they detected ASF outbreaks almost 8 weeks earlier than active slaughterhouse- and network-based surveillance.

Author Summary

1. The lay summary is concise and effectively highlights the real-world relevance.

Thank you for this comment.

2. The statement “It is not clear which contact type has the highest probability of transmission...” is appropriately answered in the subsequent text, but you might consider a more direct statement of your findings there.

We have updated the text to highlight the knowledge gap to lead into our model description. Updated text states: However, the relative contribution of these different contact types to disease spread remains unclear. Likewise, the effectiveness of existing and proposed surveillance programs is uncertain. Once an outbreak begins to spread, detecting it quickly is vital. We built a model based on data from Swiss pig holdings and transports. We simulated the spread of three infectious pathogens within and between farms, based on their different types of contact. The model simulations showed that the direct transport of pigs and geographic proximity were the main pathways of between-farm spread. We also found that surveillance programs based on farmer reporting were the most effective.

Introduction

1. The motivation for comparing ASF, PRRS, and APP is clear, but you may want to highlight more systematically why these three were chosen as representative of distinct modes of transmission or clinical severity.

We have expanded the sentence which introduces these diseases to include: `` — which represent distinct transmission characteristics and endemicity levels”. . The rationale for selecting ASF, PRRS, and APP, and their distinct modes of transmission and clinical severity, is further discussed in detail in Section 2.1 of the Materials and Methods. Additionally, Table 1 provides a side-by-side comparison of these pathogens.

2. The reference to historical data (Figure 1) is helpful to show the Swiss context. Ensuring that the figure is easy to read and interpret is important. You could consider highlighting regions with prior cases more visibly (e.g., color-coding or labeling hotspots).

We have already color-coded the regions in Figure 1 to highlight areas with prior cases. We cannot explicitly label areas with prior cases due privacy issues.

3. The hypotheses are generally well-articulated, but you might specify if you hypothesized that direct pig transports would be the most critical path or if you expected major differences in detection times among the three pathogens.

We expected the mode of transmission of the pathogen to have an impact on the performance of the surveillance strategies, and correspondingly the detection time, as stated in the following sentence in the Introduction: We hypothesize that pathogens with distinct transmission dynamics result in different performance of surveillance measures. However, we did not a priori specifically hypothesize that direct pig transports would be the most critical pathway for all pathogens.

Materials and Methods

1. Data Sources (Section 2.2)

· The coverage of TVD, trader data, and AGIS is appropriate. The detail about not having the full sequence of truck stops for all transports is a key limitation. The authors do note this, but it merits repeated emphasis, given its impact on certain contacts (tour-based).

We state in the data description that “Since these two trader companies do not service all farms in Switzerland, we do not have the truck number, collection time, nor delivery time for all records of the TVD data.” We provide the exact coverage of 15%. All of section 2.10 is related to the shortcomings of the missing 85% and the sensitivity analysis that we had done. We also discuss this shortcoming in the Discussion section with an entire paragraph, starting with:

Due to the limited published estimates on transport tour transmission rates of direct and indirect truck sharing and truck fomites, setting the parameter values for our model was difficult…

· It is beneficial that you combine farm attributes (farm type, pig numbers) with geographical coordinates. Clarify whether any missing data were excluded, or if imputation was performed for incomplete records.

For a limited subset of farms where precise geographic coordinates were not available, we used the centroid of the municipality's polygon as an approximation of the farm location. This imputation allowed us to retain these farms in the analysis while maintaining a reasonable approximation of their location. We have updated the manuscript in Section 2.2 Sources of Data with this additional text: For a limited subset of farms where precise geographic coordinates were not available, we used the centroid of the municipality's polygon as an approximation of the farm location. This allowed us to retain these farms in the analysis while maintaining a reasonable approximation of their location.

2. Within-Herd Model (SEIAR) (Section 2.3)

· The compartmental structure is clearly described.

Thank you for this comment.

· The homogeneous mixing assumption is common but can lead to overestimation of outbreak size in large farms. A brief justification is provided; perhaps highlight any known empirical evidence that suggests frequency dependence within typical Swiss pig farms.

We agree that the choice between frequency- and density-dependent transmission can affect model outcomes, especially in the context of large farm sizes. As you suggest, we have expanded our justification to clarify why frequency-dependent transmission is appropriate in the context of Swiss pig farms in Section 2.3 Within-Farm Layer. Updated text: In addition, we assumed homogeneous mixing of pigs at the farm with frequency dependent transmission. Although density-dependent transmission has often been used for animal diseases, this is more appropriate for animals kept in open spaces such on pasture where densities can change. Pig farms in Switzerland are strictly regulated with a clearly defined maximum number of animals per space by legislation. This leads to a maximum number of pigs per barn, limiting the number of contacts for each animal, even if the number of pigs per farm increases. Hence a frequency-dependent transmission assumption is more appropriate across farm systems in Switzerland.

Original text: In addition, we assumed homogeneous mixing of pigs at the farm with frequency dependent transmission. Although density-dependent transmission has often been used for animal diseases, this is more appropriate for free-roaming animals in wide open spaces or those owned by pastoralists. Pig farms in Switzerland are more strictly regulated with larger farms having more space, limiting the number of contacts, as the number of pigs increases. Hence a frequency-dependent transmission assumption is more appropriate across farm sizes in Switzerland.

3. Between-Farm Contact Networks (Section 2.4)

· The categorization of edges (D, P, I, T, G) is thorough. The distinction among direct pig transport, shared truck contacts (direct/indirect), and truck fomites is valuable.

Thank you for this comment.

· For geographic edges, clarify why you selected a specific threshold \( k \) for distance (e.g., 5 km, 10 km). A sensitivity test for different distance cutoffs might strengthen the argument that “local spread” remains important even if the radius changes slightly.

Thank you for the suggestion to clarify the choice of the distance threshold k. We selected 2 km as the cutoff based on the literature from which we derived our geographic spread parameters. Specifically, the publication we referenced used a 2 km threshold for local spread, and we adopted this same value for consistency with prior studies.

We agree that a sensitivity analysis for different distance cutoffs could further strengthen our argument regarding the importance of local spread. However, given the scope of our study, we did not conduct a sensitivity test for varying distance thresholds. We have updated the manuscript, particularly the "Between-herd transmission" paragraph in the Parameters section, to better clarify the reasoning behind the 2 km threshold. Here is the additional text:

To be consistent with the Stegeman et al. (2002) study, we chose a geographic spread threshold (k) as described in Section 2.5 to be 2 kilometers.

4. Model Implementation and Parameters

· More detail on how the authors set the parameters for PRRS, ASF, and APP (especially for indirect truck sharing probabilities) would be useful. The references for these parameter estimates are critical for the credibility of the model.

The parameters used for PRRS, ASF, and APP are provided in Table 4 (within-farm parameters) and Table 6 (between-farm parameters), with detailed explanations in Section 2.7 (Parameters section).

Additionally, we detailed the estimates here: For the indirect truck-sharing parameters (ψ), the pathway of transmission is predominantly through urine and feces. Several studies have concluded that nasal and rectal secretions are not significant transmission drivers for either CSFv or ASFv (Dewulf et al., 2002; Pietschmann et al., 2015). For ASF, we use the same ratio of ψ to ϕ as for CSF with evidence of ASF transmission through indirect contact in recent outbreaks in Estonia (Nurmoja et al., 2020). PRRSv has been shown to spread easily through urine, nasal, and feces secretion and through truck fomites (Dee et al., 2004a; Otake et al., 2002). We therefore chose to increase the ratio of indirect truck-sharing rate, ψ, to ϕ for PRSS as compared to the ratio of ψ to ϕ for CSF to account for this increased rate of transmission. In the absence of concrete data on the relative ratio of ψ to ϕ for PRRS, we make arbitrary choice of using 1.5 as the inflation factor. In Section 2, we conduct additional sensitivity analysis to consider the impact of this choice.

· In the sensitivity analysis, the authors do a robust job exploring large multiplicative changes in tour-related parameters. This is well-executed but might benefit from additional clarity on how these factors relate to realistic upper/lower bounds from literature.

In the sensitivity analysis, we chose the upper and lower bounds for the tour-related parameters based on the order of magnitude of realistic variations in these factors, rather than directly deriving them from the literature, since there is unfortunately little literature on these parameter values. Our approach was to explore a range of values tha

---

## [Decision Letter · Decision Letter 1]

22 Jul 2025

Identifying effective surveillance measures for swine pathogens using contact networks and mathematical  modeling

PONE-D-25-03280R1

Dear Dr. Moriarty,

We’re pleased to inform you that your manuscript has been judged scientifically suitable for publication and will be formally accepted for publication once it meets all outstanding technical requirements.

Kind regards,

Charles Odilichukwu R. Okpala, PhD

Academic Editor

PLOS ONE

Additional Editor Comments (optional):

I am very satisfied with the revised manuscript. It is acceptable for publication.

Reviewers' comments:

Reviewer's Responses to Questions

**Comments to the Author**

1. If the authors have adequately addressed your comments raised in a previous round of review and you feel that this manuscript is now acceptable for publication, you may indicate that here to bypass the “Comments to the Author” section, enter your conflict of interest statement in the “Confidential to Editor” section, and submit your "Accept" recommendation.

Reviewer #1: All comments have been addressed

Reviewer #3: All comments have been addressed

2. Is the manuscript technically sound, and do the data support the conclusions?

Reviewer #1: Yes

Reviewer #3: Yes

3. Has the statistical analysis been performed appropriately and rigorously? 

Reviewer #1: Yes

Reviewer #3: Yes

4. Have the authors made all data underlying the findings in their manuscript fully available?

Reviewer #1: Yes

Reviewer #3: Yes

5. Is the manuscript presented in an intelligible fashion and written in standard English?

Reviewer #1: Yes

Reviewer #3: Yes

6. Review Comments to the Author

Reviewer #1: (No Response)

Reviewer #3: Their model outcomes give evidence of the prominent transmission pathways and surveillance measures, which could help establish programs to prevent the spread of swine infectious diseases.

7. PLOS authors have the option to publish the peer review history of their article (what does this mean?). If published, this will include your full peer review and any attached files.

Reviewer #1: **Yes: **Mohamed A. Bakheet

Reviewer #3: No

---

## [Editor Report · Acceptance letter]

PONE-D-25-03280R1

PLOS ONE

Dear Dr. Moriarty,

I'm pleased to inform you that your manuscript has been deemed suitable for publication in PLOS ONE. Congratulations! Your manuscript is now being handed over to our production team.

Kind regards,

on behalf of

Dr. Charles Odilichukwu R. Okpala

Academic Editor

PLOS ONE